# Why aphid virus retention needs more attention: Modelling aphid behaviour and virus manipulation in non-persistent plant virus transmission

**Elin K. Falla** *, **Nik J. Cunniffe**

Department of Plant Sciences, University of Cambridge, Cambridge, United Kingdom

* ekf32@cam.ac.uk

**Data Availability Statement:** Supporting source code is available online at https://github.com/elinfalla/MIP-BAR_Model.

## Abstract

Plant viruses threaten food security and are often transmitted by insect vectors. Non-persistently transmitted (NPT) plant viruses are transmitted almost exclusively by aphids. Because virions attach to the aphid's stylet (mouthparts) and are acquired and inoculated via brief epidermal probes, the aphid–virus interaction is highly transient, with a very short aphid virus retention time. Many NPT viruses manipulate their host plant's phenotype to change aphid behaviour to optimise virus transmission. Epidemiological models of this have overlooked a key feature of aphid NPT virus retention: probing or feeding on a plant causes aphids to lose the virus. Furthermore, experimental studies suggest aphids could possibly inoculate multiple healthy plants within one infective period if they do not feed. Consequences of this for virus manipulation of host plant phenotype have not been explored. Our new compartmental epidemiological model includes both behaviour-based aphid dispersal and infectivity loss rates, and the ability of infective aphids to probe multiple plants before virus loss. We use our model to explore how NPT virus-induced host phenotypes affect epidemic outcomes, comparing these results to representative previous models. We find that previous models behave fundamentally differently and underestimate the benefit of an 'attract-and-deter' phenotype, where the virus induces increased aphid attraction to infected plants but deters them from prolonged feeding. Our results also highlight the importance of characterising NPT virus retention upon the aphid during probing. Allowing for multiple infective probes increases disease incidence and the effectiveness of virus manipulation, with implications for epidemic prediction and control.

## Author summary

Plant viruses can cause devastating disease epidemics. Non-persistently transmitted viruses are almost always vectored (transmitted between plants) by aphids. Experiments show virus infection can affect whether aphids are attracted to plants (by altering how infected plants 'smell'), as well as whether aphids settle for an extended feed after a brief

**Funding:** EKF acknowledges general support from Gonville & Caius College (https://cai.cam.ac.uk) and the Department of Plant Sciences (https://www.plantsci.cam.ac.uk), both University of Cambridge, through the Frank Smart-Drummond-SBS DTP University Studentship in Botany. The funders had no role in study design, data collection and analysis, decision to publish, or preparation of the manuscript.

**Competing interests:** The authors have declared that no competing interests exist.

initial probe (by altering how infected plants 'taste'). Since virus transmission requires an individual aphid to briefly probe an infected plant followed by one or more healthy plant(s), this strongly affects disease transmission. However, most studies exploring virus epidemics do not account for how aphid feeding behaviour affects how long an aphid holds the virus for, or that an aphid could infect multiple healthy plants before losing the virus. We use mathematical modelling to dissect how these aspects of aphid feeding behaviour affect virus transmission, particularly when viruses manipulate the 'smell' and 'taste' of plants. We show how previous studies, by omitting crucial aspects of aphid feeding behaviour, underestimate how viruses can promote their own transmission. We also highlight that there are very few experimental studies exploring the number of plants an aphid with the virus can consecutively infect, which is a key parameter affecting the severity of epidemics.

## Introduction

Plant viruses make up an estimated 47% of pathogens that cause plant disease epidemics [1]. Plant viruses can be transmitted horizontally (i.e. from plant to plant) by vectors, which are typically arthropods [2]. Viruses can be classified by vector transmission type, "arguably the key epidemiological characteristic of plant viruses" [2]. This includes where the virus resides in/on the vector, whether the amount of virus increases within the vector, and for how long the vector retains the virus. Non-persistent transmission (NPT) has the shortest virus retention time on the vector, on a scale of minutes to hours [3, 4]. NPT viruses make up an estimated 42% of insect-vectored plant viruses [5], and are nearly exclusively transmitted by aphids [6], the most common plant virus vector [7]. Aphids are well-suited to be vectors as their piercing–sucking mouthparts can transmit viruses without badly damaging the plant [7]. The virus is acquired through brief initial epidermal probes of an infected plant by an aphid [8, 9], and retained on the aphid's stylet (mouthparts) [3]. However, if the aphid's stylet then penetrates beyond the epidermis to initiate feeding from the infected plant's phloem, the aphid is unlikely to retain the virus [3, 10], as the virus is lost rapidly when the aphid ejects saliva from its stylet during this process [8, 9]. Therefore, transmission of NPT viruses depends on the aphid first dispersing from an infected plant after probing, then subsequently probing an uninfected plant to inoculate it with the virus [10].

Over the past decade or so, it has been increasingly recognised that NPT plant viruses frequently alter their host plant's phenotype to influence aphid vector behaviour and virus transmission [11]. This can involve changing visual/olfactory cues to alter aphid host selection, changing palatability/quality cues to mediate feeding behaviour, and affecting aphid dispersal [11–13]. Virus manipulation of plant phenotype, also called vector preference [14, 15], was first reported for an NPT virus, cucumber mosaic virus, in 2010 [16], and has since been found in numerous experimental studies across a diverse range of NPT pathosystems, for multiple aphid vectors and hosts (see [10] for review).

For NPT viruses, broadly speaking, two different phenotypes have been found. The first, dubbed 'deceptive' [16] or 'attract-and-deter' [17], involves the virus inducing its host plant to emit volatiles to attract the vector, but also increasing production of distasteful metabolites to reduce likelihood of prolonged feeding after probing (i.e. deterrence). This virus phenotype likely increases transmission at local scales [11, 17–19], because NPT virus transmission is most efficient when aphids briefly probe an infected plant's epidermis then reject it as unpalatable and disperse to another plant [20, 21]. The other virus phenotype, 'attract-and-retain' or

just 'retain' [17, 22], invokes the opposite response to the 'deter' phenotype in the host, increasing palatability of virus-infected plants by inhibiting distasteful metabolite synthesis or increasing nutrient content, sometimes also enhancing aphid performance [23, 24]. This was previously thought to inhibit NPT virus spread [6], but may in fact encourage longer-term wider-scale spread of the virus by increasing the aphid population and promoting development of winged (alate) aphids [17, 22], since alate morphs are often produced in response to high local population densities [25].

Mathematical models have played a key role in understanding virus manipulation of plant phenotype, and knowledge of how vector dynamics affect disease spread [2], avoiding the obvious difficulties of performing field-studies [26]. The first plant-specific model to include epidemiological dynamics of the virus in the vector population, introduced by Jeger et al. [27] and extended to include vector migration by Madden et al. [4], was a linked compartmental (ordinary differential equation, ODE) model with a frequency-dependent transmission function, coupling disease spread in host plants and vectors. These models introduced a theoretical framework allowing all transmission types (NPT, semi-persistent transmission, persistent transmission) to be captured by changing model parameter values. However, this generality comes at the cost of a lack of specificity to any single transmission type.

This particularly applies to how NPT virus transmission is captured by the Madden et al. [4] model. Aphid feeding and dispersal behaviour is important to NPT virus transmission because of the transient nature of virus retention on the aphid. Despite this, the model assumes both (1) a constant aphid dispersal rate between pairs of plants in the same field, and (2) a constant rate of virus (infectivity) loss from the aphid (i.e. a constant aphid infective period; note throughout this paper we use infective/infectivity here as synonymous with the more traditionally-used term viruliferous or virus-bearing). These assumptions of constant rates are particularly restrictive when considering virus manipulation of the host, which alters aphid dispersal, landing and feeding behaviours in a way that potentially depends on the current state of any epidemic [14]. Note that the Madden et al. [4] model did not incorporate virus manipulation of host plant phenotype since the phenomenon had not been experimentally observed at the time the model was developed. However, compartmental models based on the Madden et al. [4] framework have since been commonly used to investigate this topic, although not always for NPT viruses (e.g., [28, 29]). One exception is Cunniffe et al. [30], who used the same model structure to investigate both NPT and persistently-transmitted viruses but distinguished between landing and feeding, while also relaxing the assumption of a constant aphid dispersal rate. However, those authors still assumed a constant aphid infectivity loss rate.

A different ODE model structure was devised by Donnelly et al. [17] that differentiates between aphid vector probing versus feeding behaviour and relaxes the assumption of constant aphid dispersal and infectivity loss rates. The model does this through use of a Markov chain that tracks aphid 'feeding dispersals', i.e. orientation and probing behaviours of a single aphid as it moves between successive plants until it settles for an extended feed. The infection process is modelled as a "by-product of aphid probing of plants during feeding dispersals rather than as frequency-dependent contacts between susceptible hosts and infected vectors" [17]. The number of virus transmission events per aphid feeding dispersal depends on the frequency with which vectors probe an infected host followed by a susceptible one. However, unlike in the compartmental ODE models exemplified by Madden et al. [4], aphids can only probe (and therefore infect) one susceptible plant before losing infectivity in the Donnelly et al. [17] model.

This may be a significant omission. Although it is established that prolonged feeding will very likely cause an aphid to lose the NPT virus from its stylet [3, 10], there is little experimental research investigating the potential for infective probes of consecutive susceptible plants by

an aphid, if feeding does not occur. However, the small number of experiments that have been done suggest consecutive infective probes (and hence infections) are possible. For example, Watson [31] found that when given 5 minutes to probe 8 consecutive healthy plants, 100% of infective aphids were still infective by the second plant, and 10 of the 16 (62.5%) were still infective by the fourth. More recent quantitative studies on virus population bottlenecks have found similar results, estimating anywhere from 2 to 5+ plant probes before losing infectivity [32, 33]. This strongly suggests that an assumption of only probing a single susceptible plant per aphid infective period may be too severe. This is particularly likely in the case that the plant is a nonhost of the aphid species, as experimental evidence shows aphids will then reject a plant after very few probes [3].

Therefore, previous models of NPT virus transmission can, broadly-speaking, be partitioned into two classes: compartmental ODE models with constant rates of aphid dispersal and infectivity loss that do not consider aphid behaviour (but that can account for aphids infecting multiple plants within an infective period) based on Madden et al. [4], or the more recent Markov chain model of Donnelly et al. [17] that includes behaviour-based aphid rates, but no ability for multiple infective probes. In this study, we create two models based on each of these model structures, the former we refer to as the Multiple Infective Probes (MIP) model and the latter as the Behaviour-based Aphid Rates (BAR) model. We then compare them to a novel model we introduce in this paper, the MIP-BAR model. This model combines the functionality of the MIP and BAR models into a single easily-extensible system of differential equations, and allows us to answer the following three key questions (for convenience these are referenced throughout the following as Q1, Q2 and Q3).

Q1. How does replacing constant aphid dispersal and infectivity loss rates with Behaviour-based Aphid Rates change epidemic outcomes, particularly under the 'attract-and-deter' and 'attract-and-retain' phenotypes?

Q2. How does adding Multiple Infective Probes for aphids, thereby relaxing the assumption that infective aphids lose infectivity after probing one susceptible plant, affect epidemic outcomes?

Q3. What are the implications of Behaviour-based Aphid Rates and Multiple Infective Probes for our understanding of how the 'attract-and-deter' and 'attract-and-retain' phenotypes affect aphid behaviour and virus transmission?

## Materials and methods

### Model assumptions

All models (MIP, BAR and MIP-BAR) are deterministic, within-field, within-season, non-spatial models for a single host plant species. For all models, aphid behaviour is split into 'probing' and 'feeding', the former referring to brief insertions of the aphid stylet to sample superficial tissues, including brief intracellular puncture(s) to assess host compatibility, quality etc., and the latter referring to extended probes reaching the sieve elements leading to the aphid actively feeding from the phloem [34]. Aphids are assumed to make short flights between plants within the field as they search for a plant to feed on (dubbed 'dispersals' from here) and always probe after landing on a new plant, at which point they either feed or reject the plant and move to another. Given that aphids may probe a plant multiple times before rejection or acceptance [34], a 'probe' can be taken as a probing phase: one or more probes until the decision to feed or reject occurs. The probing phase is assumed to be instantaneous, as is dispersal between plants.

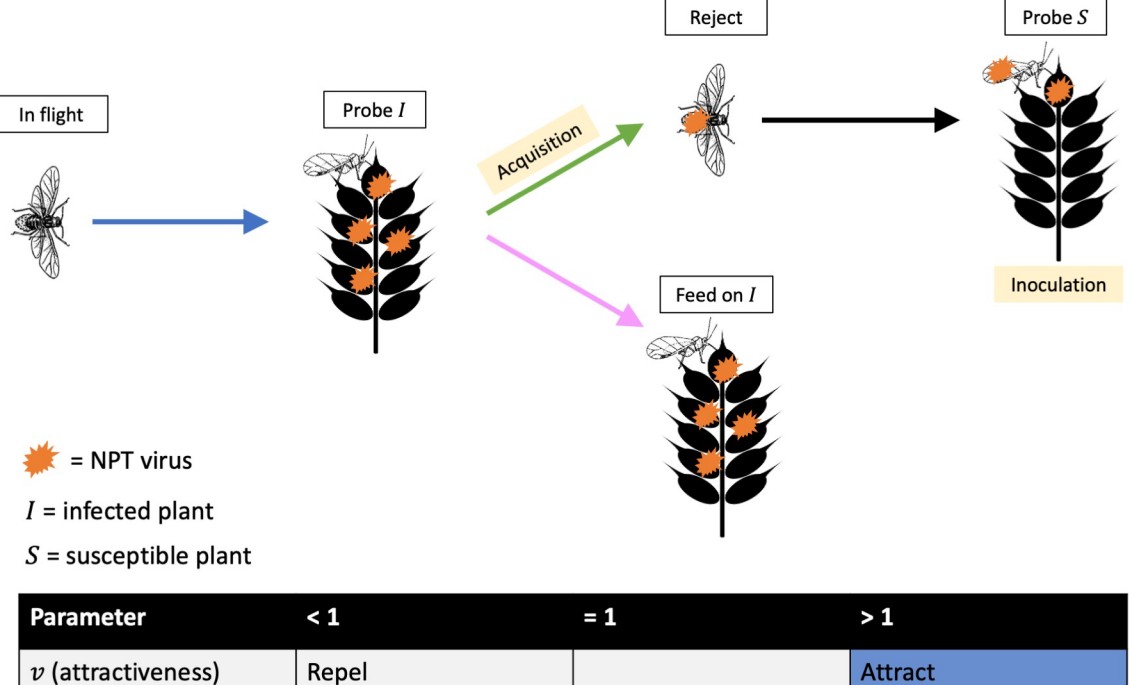

**Fig 1. The transmission process for NPT viruses, and how it can be manipulated by viruses changing infected plant attractiveness ($v$) and acceptability ($\varepsilon$).** The virus can be acquired by the aphid when it probes and does not feed on an infected plant. Inoculation can occur when the aphid then probes/feeds on a susceptible plant. Virus transmission requires both acquisition and inoculation. Potential plant phenotypes caused by viruses, and the stage of transmission process they affect, are reflected by the coloured arrows, and are split into effects on infected plant attractiveness ($v$) and acceptability ($\varepsilon$). These effects combine to create the phenotypes 'attract-and-deter' (blue and green) and 'attract-and-retain' (blue and pink). $I$ = infected plant. $S$ = susceptible plant. Note, in reality, NPT viruses are retained on an aphid's stylet (mouthparts), not within its body.

NPT virus transmission depends on these processes. As prolonged feeds cause aphids to flush the virus out of their stylet [34], an infective aphid feeding on a plant (with probability $\omega$) will always cause it to lose infectivity. Similarly, when an uninfective aphid lands on and probes an infected plant, in order for virus acquisition to occur (with probability $a$) the aphid must reject (i.e. not feed on) it. This occurs with probability $(1 - \omega)$. For NPT viruses, the aphid is immediately infective with no latent stage. For virus inoculation into a susceptible plant (which occurs with probability $b$), whether or not the aphid feeds is not important as the virus is transmitted in the probing stage [3]. However, it is important in the MIP-BAR model to determine whether the aphid retains its infectivity (see Section Model structures; MIP-BAR model).

All of the models track components of virus-induced plant phenotypes that manipulate aphid behaviour. Following Donnelly et al. [17] and Cunniffe et al. [30], this is split into infected plant attractiveness ($v$), and infected plant acceptability ($\varepsilon$) (Fig 1). The first, virus-induced infected plant attractiveness, $v$, biases aphids' landing on infected plants compared to susceptible ones ($v > 1$ = bias towards infected plants (attraction); $v < 1$ = bias away from infected plants (repulsion)). The second parameter, virus-induced infected plant acceptability, $\varepsilon$, alters the probability that an aphid will accept and feed on an infected plant after probing it, i.e. is multiplied by $\omega$ ($\varepsilon > 1$ = bias towards acceptance (retention); $\varepsilon < 1$ = bias

towards rejection (deterrence)). A value of $v = 1$ or $\varepsilon = 1$ means the virus is not manipulating plant attractiveness or acceptability, respectively. Therefore, the 'attract-and-deter' phenotype corresponds to $v > 1, \varepsilon < 1$, and the 'attract-and-retain' phenotype corresponds to $v > 1, \varepsilon > 1$ (Fig 1).

For simplicity, in all models the aphid population is assumed to be constant and to consist only of alate (winged) individuals, with no migration. For ease of comparison between the models, we also assume no aphid birth or death. These assumptions have little effect on the transmission dynamics due to the brevity of an aphid's infective period (hours) compared to its lifetime (days/weeks); see Section Default parameterisation. In addition, plants are assumed to be immediately infectious, with no latent period. The plant population is also assumed to be constant; plants can die and are immediately replaced by a susceptible plant.

## Model structures

We compare models created using two previous NPT virus model structures to our new MIP-BAR model (Table 1). The Multiple Infective Probes (MIP) model is a linked compartmental model originally created by Madden et al. [4], that has (dependent on parameterisation) the ability for infective aphids to remain infective for probing/feeding on more than one susceptible plant, i.e. infective aphids can transmit the virus to multiple plants in one infective period. The Behaviour-based Aphid Rates (BAR) model is an aphid behaviour Markov chain model created by Donnelly et al. [17] in which both (1) the number of plants an aphid visits after initiating a 'feeding dispersal' before feeding and (2) the rate that an infective aphid loses infectivity, emerge dynamically over the course of an epidemic, dependent on aphids' probing and feeding behaviour. Our novel model, the MIP-BAR model therefore combines elements of functionality from both the MIP and the BAR models. The parameters of the models are given in Table 2. Where multiple models have a parameter representing the same entity or biological phenomenon, these have been given the same symbol.

**Multiple Infective Probes (MIP) model.**   The MIP model updates Madden et al. [4] in two ways (Fig 2a). First, it contains four rather than the original seven compartments: a Susceptible-Infected (SI) model for both the plants ($S$ = susceptible/healthy, $I$ = infected) and for the aphids ($X$ = uninfective, $Z$ = infective). We omitted the Exposed (latent period) and Removed plant compartments as well as the Exposed vector compartment, which is not relevant for NPT viruses. The second change is the addition of virus manipulation of plant phenotype. To be able to add this functionality, the model needed to be able to differentiate between probing and feeding [30]. We therefore added $\omega$, the probability of an aphid feeding on a healthy plant, to the model, in order to ensure virus acquisition only occurred when the aphid does not feed (see Section Model assumptions). This allowed us to then add in virus manipulation via the parameters $v$, infected plant attractiveness, and $\varepsilon$, infected plant acceptability

**Table 1. Features and limitations of our representative previous models, in terms of aphid transmission of NPT viruses.** Our MIP-BAR model combines features of previous models for a richer representation of transmission.

| Model name/ acronym | Reference | Model name explanation | Model limitation |
| --- | --- | --- | --- |
| **M**ultiple **I**nfective **P**robes (MIP) | Madden et al. [4] | Infective aphids can probe and potentially inoculate multiple susceptible plants during one infective period | Virus transmission is not dependent on aphid feeding behaviour, just contact between aphids and plants (no BAR) |
| **B**ehaviour-based **A**phid **R**ates (BAR) | Donnelly et al. [17] | The rate that aphids lose infectivity and disperse is variable based on the epidemiological state of the system, and dependent on the aphid's probing and feeding behaviour | Aphids are guaranteed to lose infectivity after probing one susceptible plant, so cannot inoculate multiple plants in one infective period (no MIP) |

**Table 2. Table showing model parameters and their default values across the MIP, BAR and MIP-BAR models.** Not all parameters are common to all models, as indicated by the ticks and crosses.

| Parameter | In MIP? | In MIP-BAR? | In BAR? | Description | Default value (units) |
|---|---|---|---|---|---|
| $a$ | ✓ | ✓ | ✓ | Probability of virus acquisition by an aphid when probing an infected plant | 0.5 |
| $b$ | ✓ | ✓ | ✓ | Probability of virus inoculation by an infective aphid when probing a susceptible plant | 0.5 |
| $H$ | ✓ | ✓ | ✓ | Total number of plants, $I + S$ | 10000 |
| $A$ | ✓ | ✓ | ✓ | Total number of aphids, $X + Z$ | 500 |
| $\omega$ | ✓ | ✓ | ✓ | Probability of an aphid feeding on a susceptible plant | 0.2 |
| $\varepsilon$ | ✓ | ✓ | ✓ | Virus-induced infected plant acceptability (affects aphid feeding probability; see Section Model assumptions) | 1 |
| $\nu$ | ✓ | ✓ | ✓ | Virus-induced infected plant attractiveness (affects aphid landing probability; see Section Model assumptions) | 1 |
| $\Gamma$ | ✓ | ✓ | ✓ | Rate of plant death and re-planting | 0.03 (days$^{-1}$) |
| $\eta$ | ✗ | ✓ | ✗ | Average duration that an aphid feeds on a plant. Used to calculate the dispersal rate, $\widetilde{\phi}$. Default value gives $I/H$ equilibrium = 0.5 | 0.8333 (days) |
| $\rho$ | ✗ | ✓ | ✗ | Probability that an aphid loses the virus (i.e. its infectivity) upon probing a plant | 1 |
| $\widetilde{\phi}$ | ✗ | ✓ | ✗ | Aphid dispersal rate: average number of dispersal flights between plants per aphid per day. This is an expression based on aphid behaviour, that varies with $I$ (Eq 14). Default value calculated using parameters at default values and $I/H$ = 0.5 | 6.00024 (days$^{-1}$) |
| $\widetilde{\tau}$ | ✗ | ✓ | ✗ | Aphid infectivity loss rate: rate an infective aphid loses the virus per day. This is an expression based on aphid behaviour, that varies with $I$ (Eq 20). Default value calculated using parameters at default values and $I/H$ = 0.5 | 4.8 (days$^{-1}$) |
| $\phi$ | ✓ | ✗ | ✗ | Aphid dispersal rate: average number of dispersal flights between plants per aphid per day. Default value matched to $\widetilde{\phi}$ | 6.00024 (days$^{-1}$) |
| $\tau$ | ✓ | ✗ | ✗ | Aphid infectivity loss rate: rate an infective aphid loses the virus per day. Default value matched to $\widetilde{\tau}$ | 4.8 (days$^{-1}$) |
| $\theta$ | ✗ | ✗ | ✓ | Aphid feeding dispersal initiation rate: number of feeding dispersals initiated per aphid per day, where a 'feeding dispersal' is the process from initially taking flight to feeding on a plant, potentially with other plant visits in between. Default value calculated as $1/\eta$ | 0.64 (days$^{-1}$) |
| **Initial condition** | | | | | |
| $I(0)$ | ✓ | ✓ | ✓ | Initial number of infected plants ($I$) | 1 |
| $Z(0)$ | ✓ | ✓ | ✗ | Initial number of infective aphids ($Z$) | 0 |

(Fig 1), resulting in the model in Eqs 1–4. All parameters are defined in Table 2.

$$\frac{dS}{dt} = \Gamma I - \phi b Z \frac{S}{S + \nu I},$$ (1)

$$\frac{dI}{dt} = \phi b Z \frac{S}{S + \nu I} - \Gamma I,$$ (2)

$$\frac{dX}{dt} = \tau Z - \phi a (1 - \varepsilon \omega) X \frac{\nu I}{S + \nu I},$$ (3)

$$\frac{dZ}{dt} = \phi a (1 - \varepsilon \omega) X \frac{\nu I}{S + \nu I} - \tau Z.$$ (4)

As we assume the total number of plants, $H$, and the total number of aphids, $A$, are both constant, the system can readily be simplified to consist of only two equations, where $S = H - I$

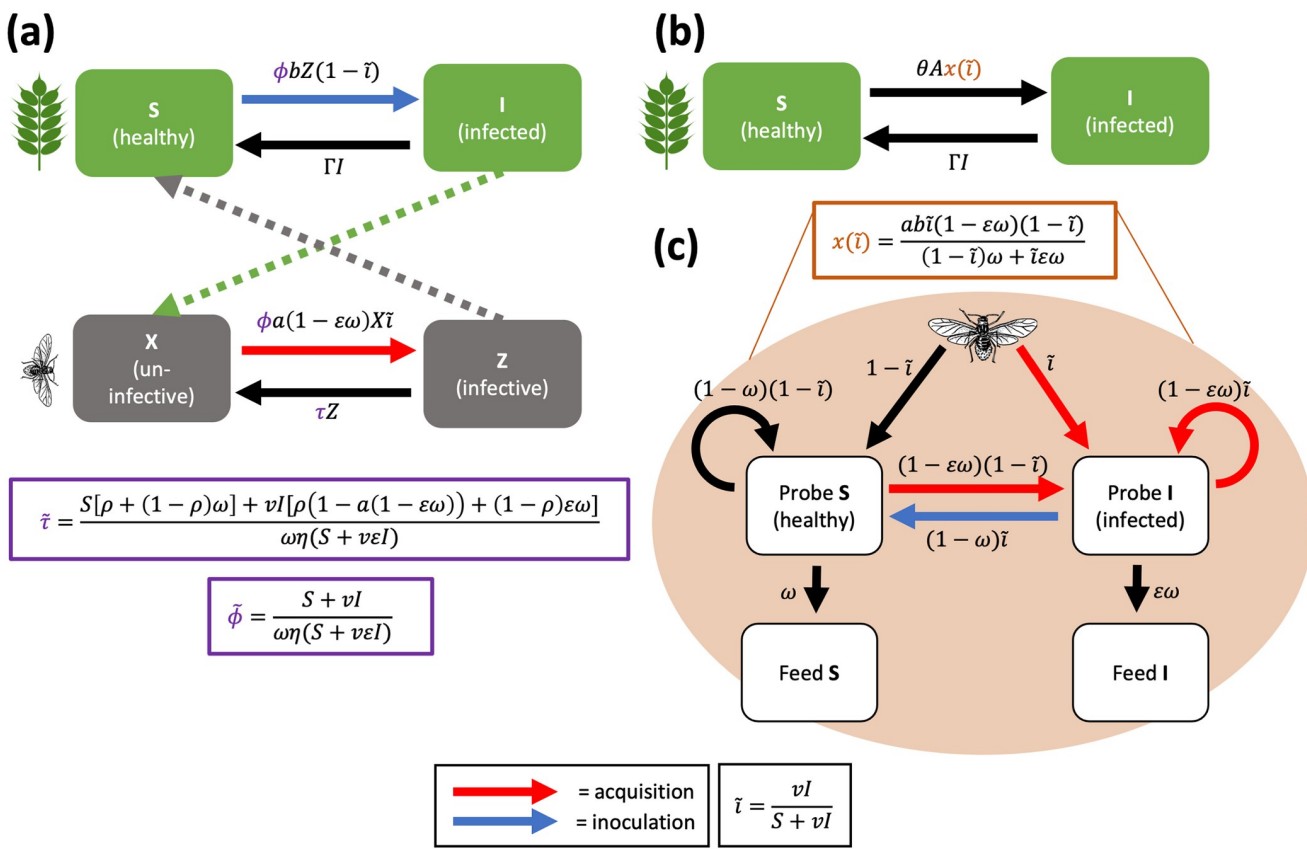

**Fig 2. Multiple Infective Probes (MIP), Behaviour-based Aphid Rated (BAR) and MIP-BAR models.** *(a)* MIP and MIP-BAR model structures. The MIP model (Eqs 1–4 in main text) has the purple parameters ($\phi$, aphid dispersal rate and $\tau$, aphid infectivity loss rate) as constants. The MIP-BAR model (Eqs 24 and 25 in main text) uses the aphid behaviour-based expressions $\widetilde{\phi}$ and $\widetilde{\tau}$ as shown in the purple boxes. *(b & c)* BAR model structure. *(b)* Compartmental model structure (Eqs 11 and 12 in main text). *(c)* Aphid feeding dispersal Markov chain as introduced by Donnelly et al. [17], the expected value of which, $x(\widetilde{i})$, is fed into the model structure in *(b)*. For all panels, red arrows = virus acquisition by uninfective aphids, blue arrows = plant inoculation by infective aphids. Virus transmission consists of acquisition and inoculation. $\widetilde{i}$ = the proportion of infected plants weighted by the infected plant attractiveness parameter, $v$. For definitions of all parameters, see Table 2.

and $X = A - Z$ (Eqs 5 and 6).

$$\frac{dI}{dt} = \phi b Z \frac{H - I}{H - I + vI} - \Gamma I, \tag{5}$$

$$\frac{dZ}{dt} = \phi a (1 - \varepsilon \omega)(A - Z)\frac{vI}{H - I + vI} - \tau Z. \tag{6}$$

**Behaviour-based Aphid Rates (BAR) model.** The BAR model is a version of the deterministic model by Donnelly et al. [17], which assumes constant plant host and aphid populations, and aphid infection dynamics always at equilibrium. It is a two state system, tracking the

rates of change of susceptible and infected plants (Fig 2b). The model is (see also Table 2)

$$\frac{dS}{dt} = \Gamma I - \theta A x(\widetilde{i}, \widetilde{s}),$$ (7)

$$\frac{dI}{dt} = \theta A x(\widetilde{i}, \widetilde{s}) - \Gamma I,$$ (8)

where

$$x(\widetilde{i}, \widetilde{s}) \quad = \frac{\widetilde{i} a (1 - \varepsilon \omega) \widetilde{s} b}{\widetilde{s} \omega + \widetilde{i} \varepsilon \omega}.$$ (9)

The expression $x(\widetilde{i}, \widetilde{s})$ (Eq 9) determines the virus transmission dynamics. It is defined as the mean number of transmissions per 'feeding dispersal', where $x(\widetilde{i}, \widetilde{s})$ is calculated by analysing a Markov chain of an aphid's landing, probing and feeding behaviour, and therefore changes over the course of an epidemic (Fig 2c; see also [17]). It can be understood intuitively as the probability of virus transmission occurring (i.e. virus acquisition then inoculation; $\widetilde{i} a (1 - \varepsilon \omega) \widetilde{s} b$), multiplied by the expected number of plants an aphid will visit within one 'feeding dispersal' $(1/(\widetilde{s} \omega + \widetilde{i} \varepsilon \omega))$. Here, $\widetilde{s} \omega + \widetilde{i} \varepsilon \omega$ is the probability of an aphid feeding; feeding marks the end of a 'feeding dispersal'. The weighted frequencies of infected and susceptible plants

$$\widetilde{i} = \frac{vI}{S + vI} \quad \text{and} \quad \widetilde{s} = \frac{S}{S + vI},$$ (10)

account for aphid landing bias due to virus manipulation of plant attractiveness.

Since the number of plants, $H = S + I$, is constant, the system can be simplified to a one-equation system using $\widetilde{s} = 1 - \widetilde{i}$ (where $x(\widetilde{i}, \widetilde{s})$ can be written instead as $x(\widetilde{i})$):

$$\frac{dI}{dt} = \theta A x(\widetilde{i}) - \Gamma I,$$ (11)

where

$$x(\widetilde{i}) = \frac{a b \widetilde{i} (1 - \varepsilon \omega)(1 - \widetilde{i})}{(1 - \widetilde{i}) \omega + \widetilde{i} \varepsilon \omega}.$$ (12)

**MIP-BAR model.** Our MIP-BAR model is structurally identical to the MIP model, with 4 compartments/ODEs. To give the model Behaviour-based Aphid Rates (BAR) functionality, we replaced the constant dispersal rate ($\phi$) and constant aphid infectivity loss rate ($\tau$) of the MIP model with expressions based on aphid landing, probing and feeding behaviour, emulating the BAR model. We call these expressions $\widetilde{\phi}$ and $\widetilde{\tau}$. Then, to add back in the MIP functionality missing from the BAR model, we added in a new parameter, $\rho$, to the aphid infectivity loss rate, where $\rho$ is the probability an infective aphid loses infectivity when probing a plant ($\rho = 1$ in the BAR model).

**Aphid dispersal rate, $\widetilde{\phi}$.** To make aphid dispersal rate variable and based on the aphid's behaviour we follow Cunniffe et al. [30] and define a new parameter, $\eta$, the average length of time an aphid feeds on a plant (days). Assuming that probing and flight are instantaneous,

Cunniffe et al. [30] calculate the average length of time spent on a plant to then be

$$\frac{\omega\eta(S + v\varepsilon I)}{S + vI}. \tag{13}$$

The average dispersal rate, $\widetilde{\phi}$, is the inverse of the average length of time spent on a plant by an aphid, leading to a final dispersal rate of

$$\widetilde{\phi} = \frac{S + vI}{\omega\eta(S + v\varepsilon I)}. \tag{14}$$

**Aphid loss of infectivity rate, $\widetilde{\tau}$.**   The behaviour of the aphid is defined by the probabilities of the aphid's possible landing, probing and feeding actions once it has become infective and dispersed (taken flight), that lead to it either losing or retaining its infectivity (Fig 3). This follows the logic of the BAR model, but with the addition of $\rho$, the probability an infective aphid loses the virus while probing. We then define aphid infectivity loss rate, $\widetilde{\tau}$, as the rate of virus loss rather than retention from one dispersal flight, i.e. the probability of moving along one of the sets of red arrows in Fig 3, multiplied by the dispersal rate, $\widetilde{\phi}$ (Eq 14).

The expression for $\widetilde{\tau}$ is therefore comprised of the probability of virus loss if the aphid lands on a susceptible plant ($\widetilde{\tau}_S$) summed with the probability of virus loss if the aphid lands on an

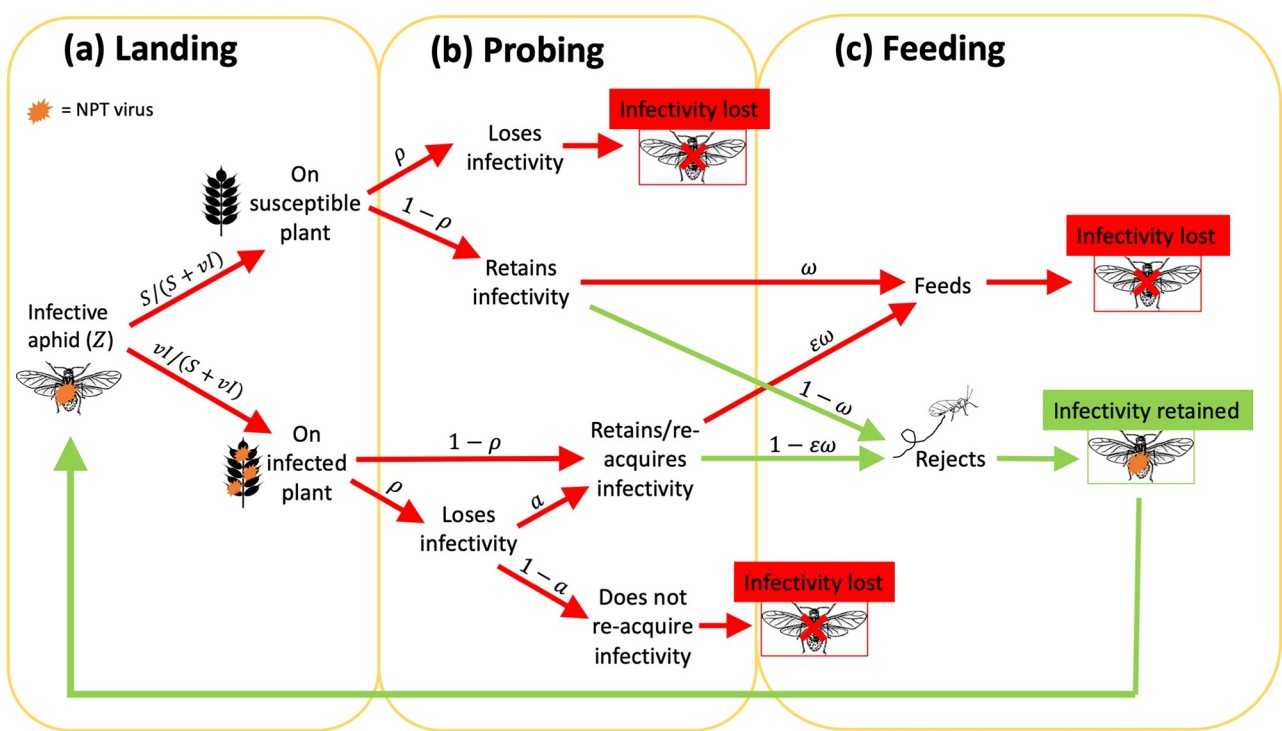

**Fig 3. Probability tree of aphid landing, probing and feeding behaviour upon which the expression for $\widetilde{\tau}$, the rate of aphid infectivity loss, in the MIP-BAR model is based.** It gives probabilities for each stage of the process (from left to right) from when an infective aphid disperses (with rate $\widetilde{\phi}$, Eq 14) through *(a)* landing, *(b)* probing and *(c)* feeding on or rejecting a new plant, which results in the aphid either maintaining or losing the virus (i.e. its infectivity). Red arrows = paths that lead to the aphid losing infectivity (which when combined create $\widetilde{\tau}$, Eq 20). Green arrows = paths that lead to infectivity retention by the aphid. $S$ = number of susceptible plants. $I$ = number of infected plants. $\rho$ = probability of infectivity loss from probing. $a$ = probability of virus acquisition from probing an infected plant. $\omega$ = probability of feeding on a plant. $v$ = degree of virus-induced plant attractiveness. $\varepsilon$ = degree of virus-induced plant acceptability.

infected plant ($\widetilde{\tau}_I$) plant, all multiplied by $\widetilde{\phi}$. The infective aphid will land on a susceptible plant with probability $S/(S + vI)$ and on an infected plant with probability $vI/(S + vI)$, accounting for the infected plant attractiveness, $v$.

If the aphid lands on a susceptible plant, probing will cause loss of infectivity with probability $\rho$. If probing does not cause infectivity loss (probability $1 - \rho$), then infectivity can still be lost if the aphid feeds (with probability $\omega$). This results in the following expression for infectivity loss probability on a susceptible ($S$) plant, $\widetilde{\tau}_S$:

$$\widetilde{\tau}_S = \frac{S}{S + vI}[\rho + (1 - \rho)\omega]. \tag{15}$$

If the aphid lands on an infected plant there is added complexity arising from the fact that, if the aphid loses the virus when probing (with probability $\rho$), it can still be regained from the infected plant itself with probability $a$. Additionally, the probability that the aphid feeds is weighted by the virus-induced infected plant acceptability parameter, $\varepsilon$. Infectivity loss therefore occurs under any of the following scenarios (Fig 3):

- The aphid loses infectivity while probing and does not regain it: $\rho(1 - a)$

- The aphid loses infectivity while probing and regains it, but then feeds on the plant: $\rho a \varepsilon \omega$

- The aphid does not lose infectivity while probing, but then feeds on the plant: $(1 - \rho)\varepsilon\omega$

The expression for infectivity loss probability on an infected ($I$) plant, $\widetilde{\tau}_I$, is therefore:

$$\widetilde{\tau}_I = \frac{vI}{S + vI}[\rho(1 - a + a\varepsilon\omega) + (1 - \rho)\varepsilon\omega], \tag{16}$$

$$= \frac{vI}{S + vI}[\rho(1 - a(1 - \varepsilon\omega)) + (1 - \rho)\varepsilon\omega]. \tag{17}$$

The rate of aphid infectivity loss when landing on any plant, $\widetilde{\tau}$, can therefore be calculated as:

$$\widetilde{\tau} = \widetilde{\phi}(\widetilde{\tau}_S + \widetilde{\tau}_I), \tag{18}$$

$$\widetilde{\tau} = \widetilde{\phi}\left\{\frac{S}{S + vI}[\rho + (1 - \rho)\omega] + \frac{vI}{S + vI}[\rho(1 - a(1 - \varepsilon\omega)) + (1 - \rho)\varepsilon\omega]\right\}. \tag{19}$$

This can be further simplified and the expression for $\widetilde{\phi}$ (Eq 14) can be substituted in, to give a final expression for $\widetilde{\tau}$

$$\widetilde{\tau} = \frac{S[\rho + (1 - \rho)\omega] + vI[\rho(1 - a(1 - \varepsilon\omega)) + (1 - \rho)\varepsilon\omega]}{\omega\eta(S + v\varepsilon I)}. \tag{20}$$

**Final model.** Using the derived expressions for the aphid dispersal and infectivity loss rate, the final MIP-BAR model is the same as the MIP model (Eqs 1–4), but with $\phi$ replaced by the expression $\widetilde{\phi}$ (Eq 14) and $\tau$ replaced by the expression $\widetilde{\tau}$ (Eq 20). As aphid and plant populations are again constant, the model can be reduced to a system of two equations using $S = H - I$ and $X = A - Z$, shown in Eqs 21 and 22. All parameter definitions are in Table 2. The

model is

$$\frac{dI}{dt} = \widetilde{\phi} b Z \frac{H-I}{H-I+vI} - \Gamma I,$$

(21)

$$\frac{dZ}{dt} = \widetilde{\phi} a (1-\varepsilon\omega)(A-Z)\frac{vI}{H-I+vI} - \widetilde{\tau} Z,$$

(22)

where

$$\widetilde{\tau} = \frac{(H-I)[\rho+(1-\rho)\omega] + vI[\rho(1-a(1-\varepsilon\omega))+(1-\rho)\varepsilon\omega]}{\omega\eta(H-I+v\varepsilon I)},$$

(23)

and

$$\widetilde{\phi} = \frac{H-I+vI}{\omega\eta(H-I+v\varepsilon I)}.$$

Or, in full, substituting in the expressions for $\widetilde{\phi}$ and $\widetilde{\tau}$, the $H - I + vI$ terms cancel out in the plant and aphid infection terms, with

$$\frac{dI}{dt} = \frac{(H-I)bZ}{\omega\eta(H-I+v\varepsilon I)} - \Gamma I,$$

(24)

$$\frac{dZ}{dt} = \frac{vIa(1-\varepsilon\omega)(A-Z)}{\omega\eta(H-I+v\varepsilon I)} - \frac{(H-I)[\rho+(1-\rho)\omega] + vI[\rho(1-a(1-\varepsilon\omega))+(1-\rho)\varepsilon\omega]}{\omega\eta(H-I+v\varepsilon I)} Z.$$ (25)

## Parameterisation and model comparison

**Default parameterisation.**   The default parameterisation for the models is intended to be biologically plausible for an NPT virus while also matching the models' disease incidence at 50% of plants infected ($I/H = 0.5$) (Table 2). We use the equilibrium proportion of infected plants as the measure for disease incidence. First, all common parameters between models were set. Following Donnelly et al. [17], we assume $a = b = 0.5$ and $\omega = 0.2$. For simplicity, by default we assume that $v = \varepsilon = 1$, i.e. no virus manipulation of plant phenotype. We assumed $H = 10,000$ and $A = 500$ (1 dispersing aphid per 20 plants, consistent with a newly colonised crop), and $\Gamma = 0.03$ days$^{-1}$, so plants are infected for 33.3 days on average. The other parameters not common between the models were then set to give an incidence of 0.5. For the MIP-BAR model, for simplicity we set $\rho = 1$ by default, so aphids always lose infectivity after probing one susceptible plant. Then $\eta$, the average length of an aphid feed on a plant, was altered to lead to a disease incidence of 0.5, resulting in $\eta = 5/6 = 0.8333$ (4 d.p.) days. To match the BAR model to this, its parameter $\theta$, the aphid feeding dispersal initiation rate per day, was altered. As $\theta$ is equivalent to the MIP-BAR model's $1/\eta$ (see S1 Appendix), its value is $1/(5/6) = 1.2$ days$^{-1}$. For the MIP model, the default parameter values (Table 2) were used to calculate the value of $\widetilde{\phi}$ (Eq 14) and $\widetilde{\tau}$ (Eq 20) at $I = 5000$ (i.e. $I/H = 0.5$). The resulting values were used for $\phi$ and $\tau$ respectively, resulting in $\phi = 6.0002$ days$^{-1}$ (4 d.p.) and $\tau = 4.8$ days$^{-1}$. We note the values of the fitted parameters are all also biologically plausible; the resulting value for $\eta$ is comparable to Cunniffe et al. [30] and $\tau = 4$ days$^{-1}$ (i.e. average aphid infective period of 1/4 day = 6 hours) is a common value in previous models [4, 27, 30, 35, 36].

**Model comparison.**   We compare epidemic outcomes throughout the analysis using the equilibrium proportion of infected plants as a metric for the final epidemic size, referred to as

'disease incidence'. Similarly, when comparing rates dependent on $I$ ($\widetilde{\tau}$ and $\widetilde{\phi}$) across parameter values, we take the value of the rate at the equilibrium proportion of infected plants, referred to as 'rate at equilibrium'.

## Results

### Invasion threshold

The basic reproduction number, $R_0$, is defined as the number of secondary infections arising from an infected individual in a population of susceptible individuals [37]. We used the Next Generation method to calculate $R_0$ for the MIP, BAR and MIP-BAR models (see S2 Appendix), leading to

$$\text{BAR}: \quad R_0 = \frac{A}{H} \cdot \frac{\theta}{\omega} \cdot av(1 - \varepsilon\omega) \cdot b \cdot \frac{1}{\Gamma}, \tag{26}$$

$$\text{MIP}: \quad R_0^2 = \frac{A}{H} \cdot \phi^2 \cdot av(1 - \varepsilon\omega) \cdot b \cdot \frac{1}{\Gamma} \cdot \frac{1}{\tau}, \tag{27}$$

$$\text{MIP-BAR}: \quad R_0^2 = \frac{A}{H} \cdot \frac{1}{\eta\omega} \cdot av(1 - \varepsilon\omega) \cdot b \cdot \frac{1}{\Gamma} \cdot \frac{1}{\rho + (1-\rho)\omega}. \tag{28}$$

Note that for the MIP and MIP-BAR models, the Next Generation method calculates $R_0^2$ by averaging over two cycles of transmission, from plant to vector and from vector to plant [30].

The components of these expressions have intuitive biological meanings. Successive terms correspond to:

- the average number of aphids per plant in the absence of the virus

- the average contact rate between aphids and plants per unit time in which both virus acquisition and inoculation can occur

- the probability of virus acquisition by the aphid during a single visit to an infected plant

- the probability of virus inoculation by the infective aphid during a single visit to a susceptible plant

- the average infective period of the plant (time units)

- the average infective period of the aphid (time units (MIP)/number of plants (MIP-BAR); not present for BAR)

Differences between the three models lie in the representation of aphid dispersal and aphid infective period. The value of $R_0^2$ in the MIP model includes the aphid dispersal rate squared ($\phi^2$), since contact with two plants (one $I$, one $S$) is required for virus transmission. By contrast, as the aphid's infective period and resulting virus transmission are explicitly linked to dispersal in the BAR and MIP-BAR models, $R_0$ (BAR) and $R_0^2$ (MIP-BAR) are linear with respect to contact with plants. The rate of contact with plants is equal to the rate of initiating a feeding dispersal (= $\theta$ in BAR, =$1/\eta$ in MIP-BAR) multiplied by the average number of plants visited until feeding, the inverse of the feeding probability, $1/\omega$. The MIP-BAR model $R_0^2$ differs from the BAR model's $R_0$, however, in that it includes the average aphid infective period, $1/[\rho + (1 - \rho)\omega]$, i.e. the average number of plants an aphid can visit before losing infectivity. This is the inverse of the probability of infectivity loss from probing/feeding.

Note that, when $\rho = 1$, and so when the aphid always loses infectivity following a probe, the MIP-BAR model $R_0^2$ simplifies to:

$$R_0^2(\rho = 1) = \frac{A}{H}\frac{1}{\eta\omega}av(1-\varepsilon\omega)b\frac{1}{\Gamma},\tag{29}$$

making it equivalent to $R_0$ for the BAR model. Also note that, under the default parameterisation (Table 2), $R_0$ or $R_0^2 = 2.000$ (3 d.p.) for all three models.

## The model trajectories differ under the default parameterisation

Although all three models were parameterised to reach a final disease incidence of 50% of plants infected, the disease trajectories differ to varying degrees. The BAR and MIP-BAR disease trajectories are very similar, whereas the MIP model epidemic takes off much faster (Fig 4a). The close similarity between the disease trajectories of the BAR and MIP-BAR models depends on the initial conditions in the MIP-BAR model, in particular $Z(0)$, the number of infective aphids at time = 0 (Fig 4b). This is because the BAR model tracks only $S$ (and $I$), not $Z$, the number of infective aphids (or $X$), and is in fact equivalent to the MIP-BAR model assuming the aphid infection dynamics are at equilibrium/steady state (S1 Appendix). Therefore, the closer $Z$ in the MIP-BAR model starts to the $Z$ null cline, i.e. the curve in $I$-$Z$ space along which $Z$ is not changing ($dZ/dt = 0$), the closer the two model trajectories are (Fig 4c). Under the default parameterisation, the MIP-BAR model trajectory moves very quickly to the $Z$ cline line then along it to equilibrium. Therefore, given $I(0) = 1$ (Table 2), $Z(0) = 0$ is close to the $Z$ cline. Increasing $Z(0)$ moves the initial conditions further from the $Z$ cline and therefore causes a progressively larger initial leap in $I$, moving the MIP-BAR epidemic trajectory further from the BAR one (Fig 4b—inset).

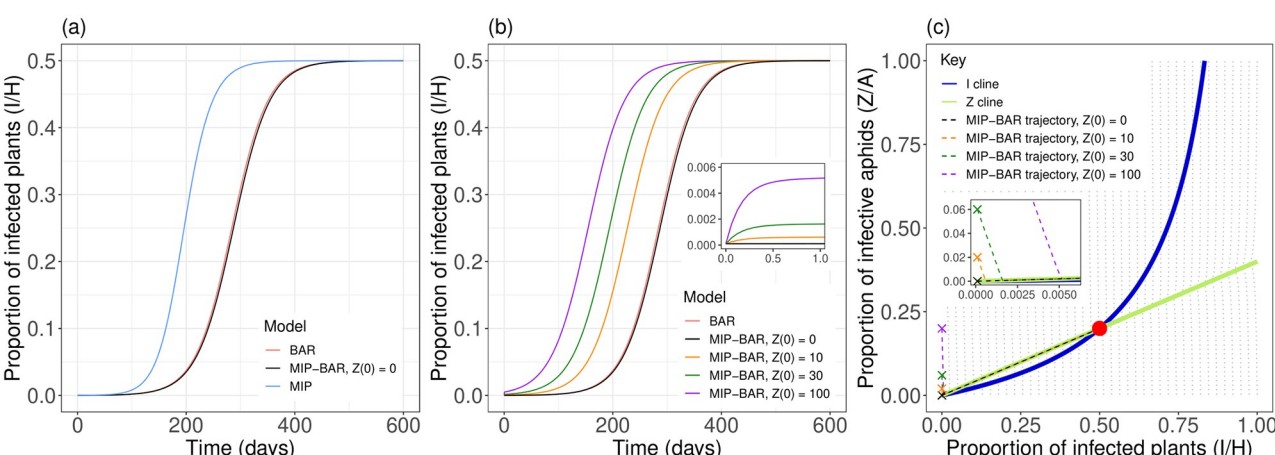

**Fig 4. MIP-BAR model is more similar to the BAR than the MIP model.** *(a)* Epidemic trajectories for MIP, BAR and MIP-BAR models using the default parameterisation (Table 2). Note it is difficult to distinguish the BAR and MIP-BAR models for this parameterisation and for the default initial condition. *(b)* Comparison of BAR trajectory to MIP-BAR trajectories at the default parameterisation, varying $Z(0)$ (the initial number of infective aphids) in the MIP-BAR model. The inset shows behaviour for small values of $t$ (up to 1 day). *(c)* $I$-$Z$ phase plane for MIP-BAR model, showing $I$ cline ($dI/dt = 0$ in Eq 24; blue) and $Z$ cline ($dZ/dt = 0$ in Eq 25; green), and epidemic trajectories from *(b)*. The inset shows a zoomed in section of the graph for small values of $Z/A$ and $I/H$, to make it clearer where the model trajectories hit the $Z$ cline.

## Q1—The Behaviour-based Aphid Rates in the MIP-BAR model change its predictions relative to the MIP model

**Varying parameters (no virus manipulation; no Multiple Infective Probes).** Under any parameterisation of the MIP-BAR model where $\rho = 1$ (i.e. without Multiple Infective Probes), the MIP-BAR and BAR models will always have the same disease incidence if the model parameters are set to the same values, and $\eta = 1/\theta$ (Fig 5, S1 Appendix). However, the differences between the MIP and MIP-BAR models are more fundamental; there is no systematic way to match their outputs. The MIP-BAR model's Behaviour-based Aphid Rates $\widetilde{\tau}$ (rate of loss of aphid infectivity, Eq 20) and $\widetilde{\phi}$ (rate of aphid dispersal, Eq 14)—which replace the equivalent constant rates $\tau$ and $\phi$ in the MIP model—are dependent on $I$, the number of infected plants, and therefore vary over the course of the epidemic in a way that the MIP model cannot replicate.

Additionally, unlike in the MIP model, altering model parameters in the MIP-BAR model changes the values of $\widetilde{\tau}$ and $\widetilde{\phi}$, with a corresponding knock-on effect on dynamics and equilibria. Model parameters can be split into three categories, in order of their degree of influence on $\widetilde{\tau}$ and $\widetilde{\phi}$: (1) those having a direct effect on both $\widetilde{\phi}$ and $\widetilde{\tau}$, (2) those having a direct effect on $\widetilde{\tau}$ only, (3) those having an indirect effect on $\widetilde{\tau}$ and $\widetilde{\phi}$ through their effects on $I$. Altering parameters with a larger influence on these Behaviour-based Aphid Rates will cause a larger difference in disease incidence between the MIP and MIP-BAR models (Fig 5).

An example of a parameter in the first category is $\omega$, the aphid feeding probability (Fig 5a, 5b and 5c), for which moving away from the default parameter value at which incidence was matched causes a larger change in disease incidence (Fig 5c) in the MIP-BAR model due to its direct influence on both the aphid infectivity loss rate (Fig 5a) and dispersal rate (Fig 5b). In the second category, $a$, the virus acquisition probability during an aphid probe, directly affects aphid infectivity loss rate (Fig 5d) as it is within the expression for $\widetilde{\tau}$, but does not affect the dispersal rate (Fig 5e). Thus altering $a$ has a much smaller effect on the resulting disease incidence, which is also more similar between the MIP and MIP-BAR models (Fig 5f). Finally, $\Gamma$, the plant death and replanting rate, is an example of a parameter that has only an indirect influence on $\widetilde{\tau}$ by affecting $I$ (Fig 5g), and once again no effect on aphid dispersal (Fig 5h), causing an even smaller change in disease incidence compared to parameters directly influencing $\widetilde{\phi}$ or $\widetilde{\tau}$ (Fig 5i).

Note that $a$ and $\Gamma$ do not have even an indirect effect on the aphid dispersal rate $\widetilde{\phi}$ via their effect on $I$, because under a no virus manipulation scenario ($v = \varepsilon = 1$), $\widetilde{\phi}$ simplifies to just $1/\omega\eta$, which is independent of $I$. Additionally, when there are no infected plants ($I = 0$), $\widetilde{\tau}$ also simplifies to $1/\omega\eta$, i.e. the infectivity loss rate becomes equal to the dispersal rate (see also S3 Appendix). This causes the kink in the $\widetilde{\tau}$ response as $\omega$ increases (Fig 5a) and the plateaus in the responses of $\widetilde{\tau}$ to $a$ and $\Gamma$ (Fig 5d and 5g).

**Varying virus manipulation parameters (no Multiple Infective Probes).** The same logic can be applied to the virus manipulation parameters: $v$, infected plant attractiveness, and $\varepsilon$, infected plant acceptability (Fig 6). Although both parameters have a direct effect on both $\widetilde{\phi}$ and $\widetilde{\tau}$ (see Eqs 14 and 20), when $\varepsilon = 1$ the aphid dispersal rate $\widetilde{\phi}$ in the MIP-BAR model simplifies to $1/\omega\eta$, so is unaffected by the value of $v$ (Fig 6b). This explains why, when $\varepsilon = 1$, the disease incidence does not differ very much between the MIP and MIP-BAR models when varying $v$ (Fig 6c), despite $v$ being present in $\widetilde{\tau}$ and causing decreased $\widetilde{\tau}$ as $v$ increases (Fig 6a). Increasing $v$ decreases $\widetilde{\tau}$ by increasing the likelihood aphids land on infected plants and prolong their infectivity. In contrast, the value of $\varepsilon$, infected plant acceptability, affects both the

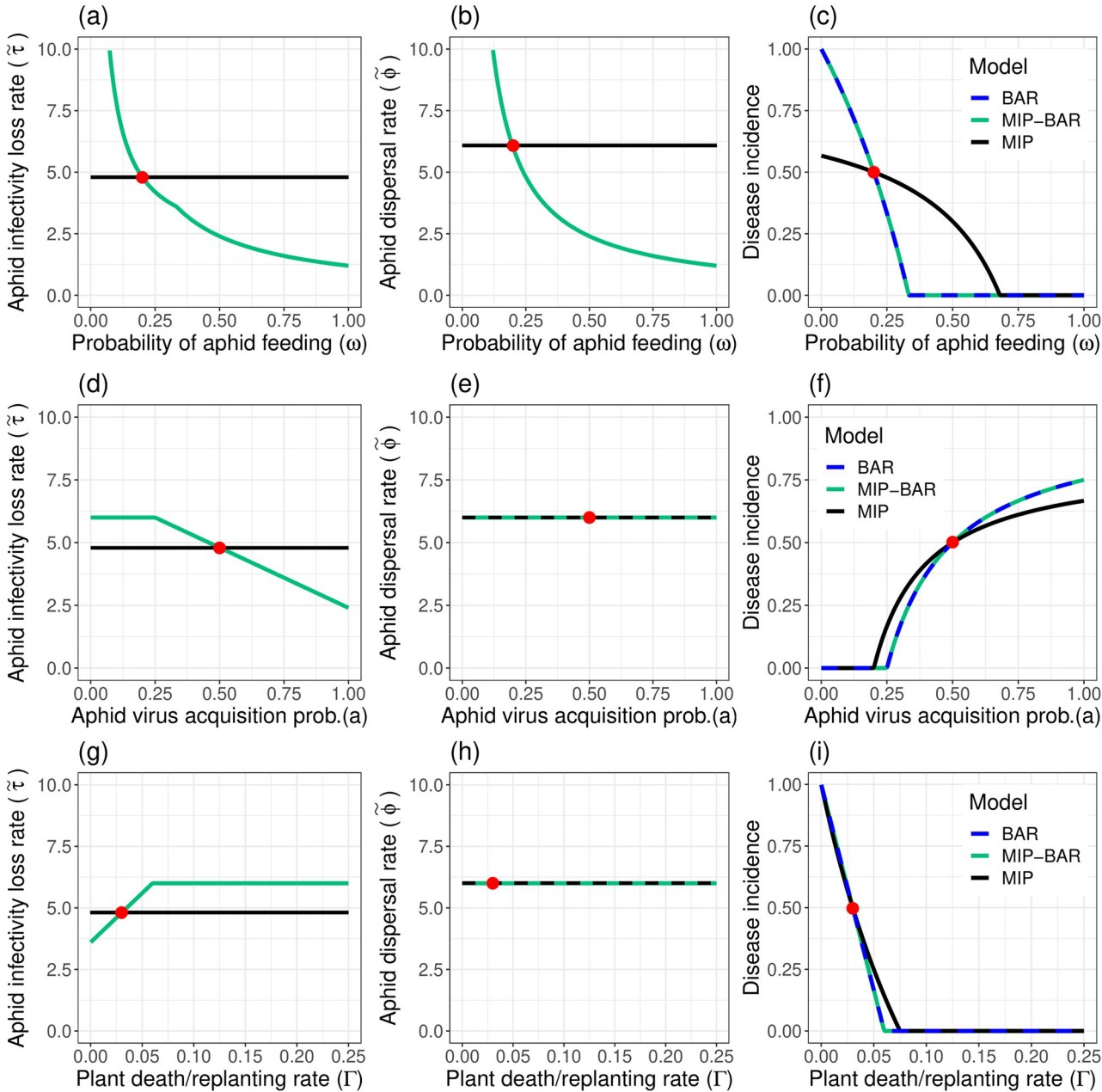

**Fig 5. The MIP and MIP-BAR model predictions vary differently across parameter values, driven by variability of behaviour-based aphid rates $\widetilde{\phi}$ and $\widetilde{\tau}$ in the MIP-BAR model.** Each column has the same plot for: $\omega$ (aphid feeding probability; top row), $a$ (aphid virus acquisition rate; middle row) and $\Gamma$ (plant death/replanting rate; bottom row) parameters, respectively. *Left (a)/(d)/(g)*: parameter versus MIP-BAR/MIP model aphid infectivity loss rate (rate at equilibrium for MIP-BAR model, calculated as in Eq 20); *Middle (b)/(e)/(h)*: parameter versus MIP-BAR/MIP model aphid dispersal rate (rate at equilibrium for MIP-BAR model, calculated as in Eq 14); *Right (c)/(f)/(i)*: parameter versus disease incidence (equilibrium $I/H$) for MIP, MIP-BAR and BAR models. For all plots, the red point signifies the default parameterisation. Apart from the parameter being altered in each graph, all parameters are at their default values in all plots (Table 2). Note that in all cases the disease incidence for the BAR model and the MIP-BAR model are identical; note further that the parameters $\widetilde{\phi}$ and $\widetilde{\tau}$ are not defined for the BAR model.

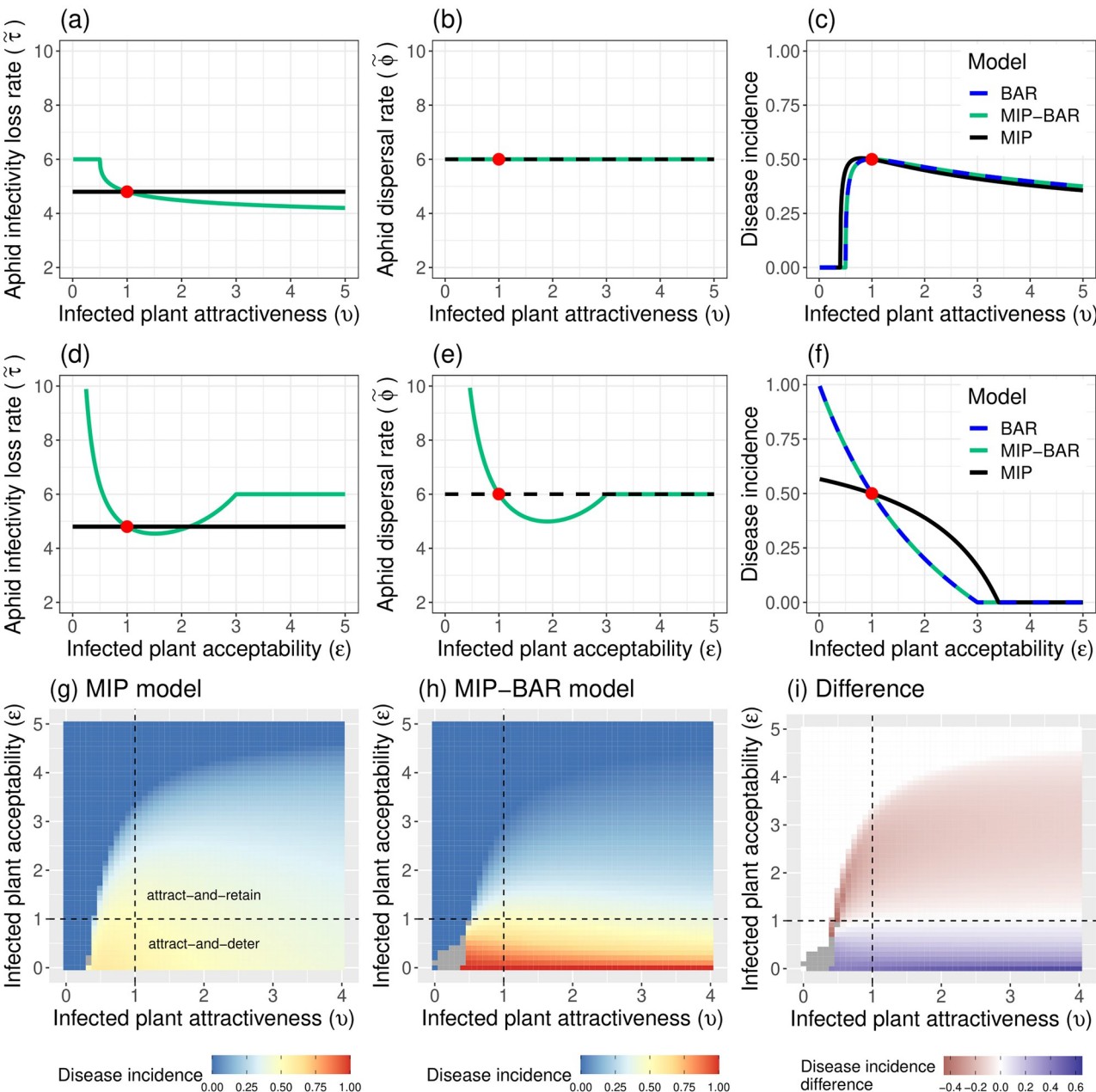

**Fig 6. MIP-BAR and MIP models respond differently to virus manipulation parameters ν (infected plant attractiveness) and ε (infected plant acceptability), leading to different predictions for effects of 'attract-and-deter' and 'attract-and-retain' phenotypes.** *(a-f)* Columns represent the same plots for the two virus manipulation parameters, ε (infected plant acceptability) and ν (infected plant attractiveness). *Left (a)/(d)*: parameter versus MIP-BAR/MIP model aphid infectivity loss rate (rate at equilibrium for MIP-BAR model, calculated as in Eq 20); *Middle (b)/(e)*: parameter versus MIP-BAR/MIP model aphid dispersal rate (rate at equilibrium for MIP-BAR model, calculated as in Eq 14); *Right (c)/(f)*: parameter versus disease incidence (equilibrium $I/H$) for MIP, MIP-BAR and BAR models. For all plots, the red dot signifies the default parameterisation. Apart from the parameter being altered in each graph, all parameters are at their default values in all plots (Table 2). Note that in all cases the disease incidence for the BAR model and the MIP-BAR model are identical; note further that the parameters $\widetilde{\phi}$ and $\widetilde{\tau}$ are not defined for the BAR model. *(g-i)* Heatmaps across ν-ε parameter space of: *(g)* MIP model disease incidence, *(h)* MIP-BAR model disease incidence, *(i)* Difference in disease incidence between MIP-BAR and MIP models (MIP-BAR—MIP). Grey = areas of multiple stable $I/H$ equilibria (or where difference cannot be calculated due to this, for *(i)*). Dotted lines are at ν = 1 and ε = 1, lines of no virus manipulation. To the right of the line of ν = 1 represents the 'attract' phenotype. Above the line ε = 1 is the 'retain' phenotype, and below is 'deter'. All parameters except ν and ε are at their default values in all plots (Table 2).

dispersal and infectivity loss rate (even when $v = 1$), by changing the probability an aphid feeds on an infected plant. Decreasing $\varepsilon$ below 1 therefore has a similar effect to decreasing $\omega$ (cf. Fig 5a, 5b and 5c), causing a large increase in the aphid dispersal rate (Fig 6e) and hence infectivity loss rate (Fig 6d). The consequence is a much more pronounced effect on disease incidence in the MIP-BAR model than the MIP model (Fig 6f). However, unlike $\omega$, $\varepsilon$ has no effect on $\widetilde{\tau}$ or $\widetilde{\phi}$ when the epidemic has died out ($I = 0$) and $\widetilde{\tau} = \widetilde{\phi} = 1/\omega\eta$ (see S3 Appendix), as $\varepsilon$ (and $v$) only relate to infected plants.

Expanding out to all of $v$-$\varepsilon$ (virus manipulation) parameter space confirms that the MIP-BAR model responds much more strongly to the value of $\varepsilon$, infected plant acceptability, than the MIP model (Fig 6g, 6h and 6i). The MIP model predicts a smaller epidemic than the MIP-BAR for $\varepsilon < 1$ ($\sim 0.2 - 0.5$ disease incidence difference) and a larger epidemic than MIP-BAR for $\varepsilon > 1$ ($\sim 0.1 - 0.2$ disease incidence difference). The two models are more concordant across values of $v$, infected plant attractiveness, however, with the MIP-BAR model showing a slightly larger epidemic increase with increasing $v$, particularly at low values of $\varepsilon$. Both models also show bi-stability between a zero and non-zero equilibrium [30] for select low values of $v$ and $\varepsilon$ (shown in grey in the figure), but the region of bistability is much larger for the MIP-BAR model.

## Q2—Multiple Infective Probes causes the MIP-BAR model to diverge from the BAR model

The results of the BAR and MIP-BAR models differ when $\rho < 1$ in the MIP-BAR model (i.e. when allowing for Multiple Infective Probes). Reducing $\rho$, the probability aphids lose infectivity from probing ($\rho = 1$ in BAR model), causes the MIP-BAR model to have a progressively larger epidemic compared to the BAR model, with progressively faster disease progression (Fig 7a). Even a small decrease in $\rho$ has a large impact, but under the default parameterisation, reducing $\rho$ to 0.5 causes $\sim 15\%$ more plants to become infected and the virus reaches 50% of the final incidence approximately twice as rapidly. A $\rho$ value of 0.5 corresponds to assuming an

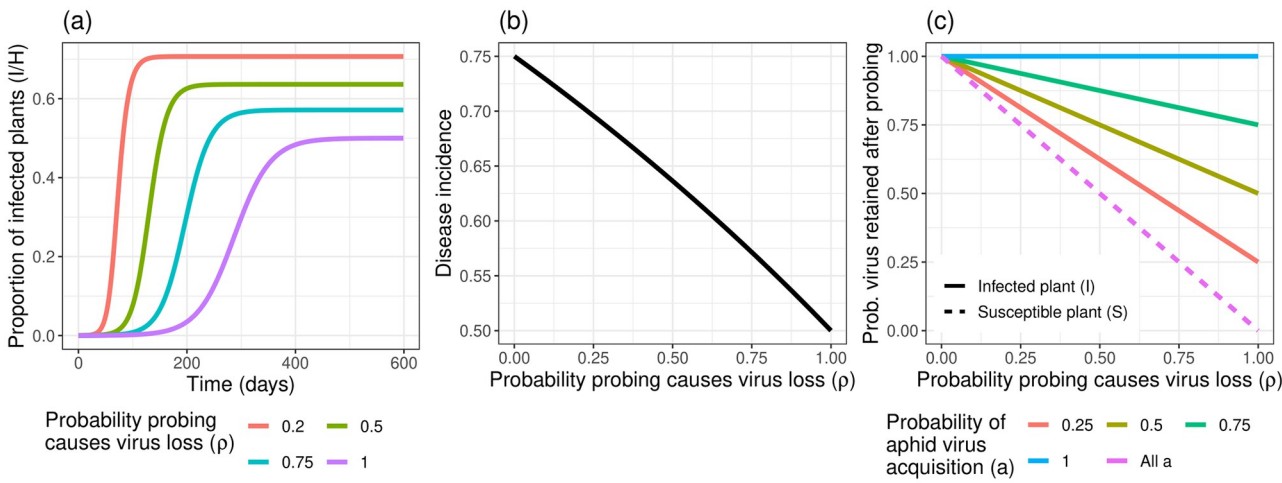

**Fig 7. Relaxing the assumption of aphids always losing infectivity from probing ($\rho = 1$) increases epidemic size in the MIP-BAR model.** (a) Epidemic trajectories (proportion of infected plants over time) for different values of $\rho$, under otherwise default parameterisation. Purple line = default parameterisation ($\rho = 1$. equivalent to the BAR model). (b) Relationship between $\rho$ and disease incidence at equilibrium. (c) Effect of $\rho$ on the probability of virus retention by the aphid after probing, for both infected and susceptible plants, for different values of $a$ (probability of aphid virus acquisition from an infected plant). When probing infected plants, the retention probability = $(1 - \rho) + \rho a$, where $a$ is the probability of virus (re-) acquisition from the plant from probing. For susceptible plants, the retention probability is always just $1 - \rho$.

infective aphid can probe on average two susceptible plants before losing infectivity rather than one. Reducing $\rho$ further to 0.2, which assumes infective aphids can probe on average 5 plants before infectivity loss, causes $\sim 20\%$ more plants to become infected than when $\rho = 1$. The relationship between $\rho$ and disease incidence is near-linear for the default parameterisation (Fig 7b).

Reducing $\rho$ increases epidemic size by changing the probability of virus retention versus loss when probing (Fig 7c). On a susceptible plant, when $\rho = 1$ there is no chance of virus retention when probing. However, on an infected plant, although the aphid will lose the virus when probing, it could potentially regain it from the infected plant itself. The probability of virus retention in this case is therefore equal to $a$, the probability of virus acquisition when probing an infected plant. As $\rho$ decreases, the probability of retaining the virus from probing increases linearly. For susceptible plants, the probability of retention is simply $1 - \rho$, but for infected plants, it is the probability of retention, $1 - \rho$, plus the probability of regaining the virus from the plant if it is lost, $\rho a$. The lower the value of $a$, the larger the effect of decreasing $\rho$ on the probability of virus retention from probing (i.e. the larger the gradient). The effect of reducing $\rho$ is therefore most pronounced for susceptible plants, for which regaining infectivity from the plant during a probe is impossible. For example, under the default parameterisation with $a = 0.5$, reducing $\rho$ to 0.5 causes retention probability to increase by 0.25 for infected plants (Fig 7c, yellow line), but by 0.5 for susceptible plants (Fig 7c, pink line).

## Q3—The MIP-BAR model gives insights into the effects of 'attract-and-deter' and 'attract-and-retain' phenotypes on aphid behaviour

We consider three indicative virus manipulation scenarios:

1. No manipulation of plant phenotype ($v = \varepsilon = 1$).

2. Attract-and-deter (exemplified by $v = 1.5, \varepsilon = 0.5$).

3. Attract-and-retain (exemplified by $v = 1.5, \varepsilon = 2$).

We distinguish cases without and with Multiple Infective Probes (i.e. $\rho = 1$ *vs.* $\rho < 1$).

**Without Multiple Infective Probes ($\rho = 1$).** Even when $\rho = 1$, our MIP-BAR model predicts different epidemic outcomes for the different virus manipulation scenarios (Fig 8a). This is driven by differences in the aphid infectivity loss rate, $\widetilde{\tau}$ (Fig 8c), aphid dispersal rate, $\widetilde{\phi}$ (Fig 8d), and aphid virus acquisition rate (Fig 8e). The aphid virus acquisition rate, $\widetilde{\phi} a(1 - \varepsilon\omega)vI/(S + vI)$, is made up of the rates at which uninfective aphids disperse ($\widetilde{\phi}$), land on infected plants ($vI/(S + vI)$) and acquire the virus (i.e. acquire the virus from probing and do not stay for a prolonged feed, $a(1 - \varepsilon\omega)$). Under no virus manipulation, the rate of aphid infectivity loss, $\widetilde{\tau}$, decreases linearly as the proportion of infected ($I$) plants increases, as aphids are more likely to land on $I$ plants and therefore more likely to prolong their infectivity. The aphid dispersal rate is constant as the epidemic progresses, because with $v = \varepsilon = 1$, $\widetilde{\phi}$ simplifies to $1/\omega\eta$, so is independent of $I$. As $\widetilde{\phi}$ is constant, under no virus manipulation aphid virus acquisition increases linearly with increasing $I$ for the same reason as $\widetilde{\tau}$ decreases: aphids become more likely to land on infected plants as an epidemic progresses.

With virus manipulation of plant phenotype, however, these trends change. The 'attract-and-deter' phenotype increases the aphid dispersal rate $\widetilde{\phi}$ as the epidemic progresses, because aphids are increasingly deterred from feeding as the proportion of $I$ plants increases. This causes both: (1) the aphid virus acquisition rate to increase at a faster rate and (2) the infectivity loss rate ($\widetilde{\tau}$) to increase slightly as the epidemic progresses, as both rates are directly

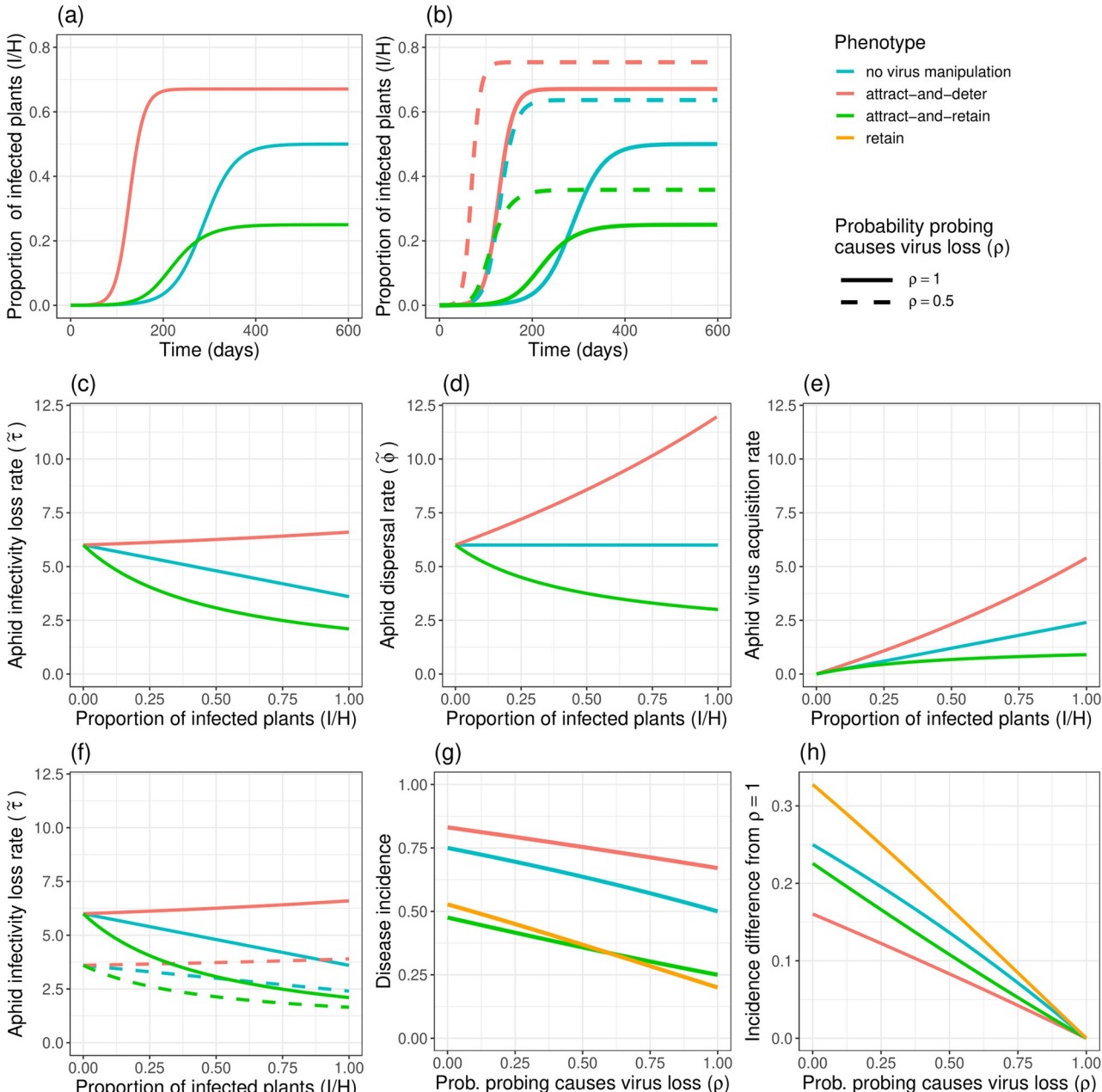

**Fig 8. The effects of the 'attract-and-deter' and 'attract-and-retain' phenotypes on epidemic size are driven by the changes they induce in aphid dispersal, $\widetilde{\phi}$, and infectivity loss, $\widetilde{\tau}$, rates, and vary to differing degrees with decreasing $\rho$.** Line colours represent virus-induced plant phenotypes: blue = no virus manipulation ($v = \varepsilon = 1$), red = 'attract-and-deter' scenario ($v = 1.5$, $\varepsilon = 0.5$), green = 'attract-and-retain' scenario ($v = 1.5$, $\varepsilon = 2$). Linetype represents the value of $\rho$, the probability of an aphid losing infectivity (the virus) when probing, for *(a)-(f)*. *(a)* Model trajectories under each plant phenotype, for $\rho = 1$. *(b)* Model trajectories under each plant phenotype, for $\rho = 1$ and $\rho = 0.5$. *(c)* Rate of aphid infectivity loss, $\widetilde{\tau}$ (Eq 20), across proportion of plants infected ($I/H$) for each phenotype. *(d)* Aphid dispersal rate, $\widetilde{\phi}$ (Eq 14), across proportion of plants infected ($I/H$), for each phenotype. *(e)* Aphid rate of virus acquisition ($\widetilde{\phi}a(1 - \varepsilon\omega)vI/(S + vI)$, the positive term in $dZ/dt$, Eq 22, but without multiplication by $X$ in order to make it a per-aphid rate) across proportion of plants infected ($I/H$), for each phenotype. *(f)* Rate of aphid infectivity loss, $\widetilde{\tau}$ (Eq 20), across proportion of plants infected ($I/H$) for each phenotype, for $\rho = 1$ and $\rho = 0.5$. *(g)* Disease incidence (equilibrium $I/H$) across values of $\rho$ for each phenotype, as well as for an additional 'retain' phenotype ($v = 1$, $\varepsilon = 2$). *(h)* Difference in disease incidence from its value at $\rho = 1$ (i.e. difference is 0 at $\rho = 1$) across values of $\rho$, for each phenotype, including the aforementioned additional 'retain' phenotype. All other parameters are at their default values.

proportional to $\widetilde{\phi}$. This results in a larger epidemic and a faster onset than the no virus manipulation scenario. Conversely, under an 'attract-and-retain' scenario the dispersal rate $\widetilde{\phi}$ decreases as the epidemic progresses, as aphids are more likely to stay for a prolonged time on plants, due to the increase in $I$ plants which are more acceptable for feeding. This also causes the aphid infectivity loss rate $\widetilde{\tau}$ to decrease much faster, and the virus acquisition rate to nearly plateau as the epidemic progresses, since aphids are dispersing less and therefore spreading the virus less. This results in a smaller epidemic with a longer onset than under a no virus manipulation scenario.

**With Multiple Infective Probes ($\rho < 1$).**    As is seen under no virus manipulation of plant phenotype ($\nu = \varepsilon = 1$), decreasing $\rho$, the probability of aphid virus loss from probing, causes a larger epidemic across both the 'attract-and-deter' and 'attract-and-retain' phenotypes (Fig 8b), which is driven by a decrease in aphid infectivity loss rate, $\widetilde{\tau}$, (i.e. increase in aphid infective period) across all values of $I$. For the 'attract-and-retain' and no virus manipulation scenarios, the decrease is more pronounced at lower $I$ (Fig 8f, green and blue lines), but for 'attract-and-deter' it is fairly consistent across values of $I$ (Fig 8f, red lines). Reducing $\rho$ also reduces the differences in aphid infectivity loss rate between the different virus manipulation phenotypes (Fig 8f). The degree of difference in disease incidence caused by reducing $\rho$ differs across the phenotypes, however. While the 'attract-and-deter' phenotype remains the phenotype causing the largest epidemic across all values of $\rho$ (Fig 8g), the difference in disease incidence caused by reducing $\rho$ is less than for 'attract-and-retain', which is in turn less than under no virus manipulation (Fig 8h). The 'retain' phenotype alone (without the 'attract' component; $\nu = 1, \varepsilon = 2$) shows the largest increase; the 'attract' phenotype ($\nu > 1$) reduces the effect of $\rho$ on epidemic size.

## Discussion

### Q1—Replacing constant aphid dispersal and infectivity loss rates with Behaviour-based Aphid Rates reveals deficiencies in previous models

Adding Behaviour-based Aphid Rates, $\widetilde{\tau}$ (aphid infectivity loss) and $\widetilde{\phi}$ (aphid dispersal rate) to the Multiple Infective Probes (MIP) model fundamentally changes its behaviour. Instead of having a fixed average length of time spent on a plant ($1/\phi$) and fixed average infective period ($1/\tau$), these time periods vary with $I$ and both directly and indirectly depend on the values of the model parameters (Fig 5). Therefore the MIP model cannot be considered a special case of the MIP-BAR model; there is no systematic way to match them. We argue, as Donnelly et al. [17] (and to an extent Cunniffe et al. [30]) did, that these Behaviour-based Aphid Rates are essential to ensuring that NPT virus transmission by aphids is accurately captured. This is particularly relevant for $\widetilde{\tau}$, the aphid loss of infectivity rate. A constant rate $\tau$ cannot capture the fact that, under a scenario with no virus manipulation of plant phenotype ($\nu = \varepsilon = 1$), the length of an aphid's infective period depends on whether it lands on primarily $S$ or $I$ plants. As the epidemic progresses and the number of $I$ plants increases, the aphid has on average a longer infective period through landing more often on $I$ plants. This is because on an $I$ plant its infectivity can be prolonged by picking up more virions from the plant during probing. A behaviour-based infectivity loss rate is therefore more representative of NPT transmission. Even at constant prevalence of infection, the MIP-BAR model also captures the unavoidable co-dependence between rate of aphid dispersal and the rate of loss of infectivity. In previous papers of NPT virus transmission using the MIP model structure [4, 27, 30, 35, 36], these are assumed to be independent. In fact, most previous models do not even make the distinction between an aphid probing versus feeding [4, 27, 35, 36]. In the MIP-BAR model, infectivity

loss rate ($\tilde{\tau}$) is directly proportional to the dispersal rate ($\tilde{\phi}$). This has a profound impact on epidemic outcomes, particularly when altering parameters that directly impact the dispersal rate, $\tilde{\phi}$ (Eq 14), as aphid dispersal is the mechanism by which virus transmission occurs (see Figs 5 and 6).

The infected plant acceptability parameter, $\varepsilon$, is one such parameter, as it changes the probability aphids feed on $I$ plants, and therefore affects the number of dispersals made per unit time. When compared to the MIP model, the MIP-BAR model has increased epidemic size at low $\varepsilon$ values and decreased epidemic size at higher $\varepsilon$ values, which occurs across all values of $\nu$ (infected plant attractiveness). This shows that when the average aphid infective period and dispersal rate are properly coupled with aphid probing and feeding behaviour, the epidemic size is much more responsive to virus manipulation-induced feeding deterrence/retention. This means that the MIP model, in not including Behaviour-based Aphid Rates, underestimates the value of the 'attract-and-deter' phenotype in increasing virus spread (Fig 6i, deter = $\varepsilon < 1$, see Fig 1). Note, very low values of $\varepsilon$ are unlikely as the aphids risk death with the very high dispersal rate it causes [17, 38].

## Q2—Relaxing the assumption that infective aphids can only probe one susceptible plant before losing infectivity increases epidemic severity

We might question whether Multiple Infective Probes are necessary. When $\rho = 1$ in the MIP-BAR model, i.e. when aphids are guaranteed to lose infectivity when probing a susceptible plant, the model is functionally very similar to the BAR model, always having the same disease incidence. Under these conditions, the only difference between the models is that the MIP-BAR model tracks aphid infection dynamics whereas the BAR model assumes aphid infection dynamics are at steady state (S1 Appendix). Therefore, the closer the initial number of infective aphids ($Z(0)$) starts to its equilibrium value for the given $I(0)$, the closer the epidemic trajectories of the two models (Fig 4). The BAR model's assumption of aphid infection dynamics at steady state implies a separation of timescales of the plant and aphid infection dynamics. Given that aphids hold the virus very transiently (minutes to hours) whereas, at least for most viruses, plants are infected for their entire lifetime [34], the BAR model simplification is likely justified in most cases.

However, the strong assumption the BAR model makes of $\rho = 1$, i.e. that infective aphids always transmit all viable virions to a single susceptible plant, is in our view not justified. Although it is established that prolonged feeding very likely causes transmission of all NPT virions from aphid to plant [3, 10], there is little experimental research (and, to our knowledge, no field experiments) investigating the potential for multiple infective plant probes by an aphid in the event that feeding does not occur. However, experiments were done some time ago which indicate consecutive inoculations are possible [31, 39]. For example, Watson [31] fed aphids for two minutes on Potato virus Y (PVY) infected plants, then moved the aphids to eight consecutive healthy plants for five minutes per plant. They found that 100% (of 16) aphids were still infective when put on the second healthy plant, i.e. 100% of aphids infected either the second plant or one of the following ones. 10 of 16 (62.5%) were still infective by the fourth plant, and 2 (12.5%) were still infective by the eighth. Note also that 15 of 16 (93.75%) aphids failed to infect at least one plant while still infective, reflecting that an infective probe(s) does not always translate to a plant inoculation. Our MIP-BAR model accounts for this too; the probability of inoculation is determined by a separate parameter $b$ (see Table 2).

Useful information can also be gleaned from more recent quantitative studies of virus population bottlenecks in NPT virus transmission. For example, Moury et al. [33] used experimental and modelling techniques to estimate the number of PVY particles that are picked up and

transmitted by aphids to a single pepper plant. They determined that the average number of probing punctures required by an aphid to release all their acquired virus particles is 4.37 to 5.25. A later study on Plum Pox Virus by Moreno et al. [32] found that aphids lose approximately half of their acquired virus particles in a single probe of a healthy plant, and that 6.9% were still infective after five intracellular punctures.

It is true that, when not manually being moved, aphids may exhaust all their infective probes on one plant. However, given evidence that most initial probes (regardless of whether the plant is a suitable host) include one brief cellular puncture [40], and that it "seems likely that the decision to fly or remain on the plant occurs shortly after a cell puncture" [41], these infective probes could plausibly occur across multiple plants. (The MIP-BAR model takes into account that aphids usually probe more than once before feeding/rejection—see Section Materials and methods; Model assumptions).

The possibility that $\rho < 1$ is important because when the assumption of $\rho = 1$ is relaxed, there are significant implications for epidemic onset speed and eventual size. For our default parameterisation at least, decreasing $\rho$ more or less linearly increases epidemic size, and also greatly increases epidemic rate of increase (Fig 7); a reduction in $\rho$ from 1 to 0.2 (meaning aphids can on average probe five plants before losing infectivity, the upper limit of what experimental studies suggest is plausible) reduces time to disease equilibrium by a factor of four. A faster disease onset in particular increases the difficulty of disease detection and control. For future epidemiological models of NPT viruses, therefore, more studies are needed on the number of plants an aphid can probe before losing all NPT virus particles acquired.

We note that the MIP functionality could, perhaps, be added to the BAR model directly using its current Markov chain structure, rather than using the compartmental model structure of the MIP-BAR model. However, relaxing the $\rho = 1$ assumption greatly complicates the Markov chain diagrammed in Fig 2c, as the infectivity of the aphid must also be tracked as well as its feeding behaviour. This results in a 7-state Markov chain, with a closed form solution for $x(\widetilde{i})$ impossible by hand, requiring numerical calculation at every time point in the epidemic and meaning analytic summaries such as $R_0$ can no longer be found in closed form (see S4 Appendix). We therefore feel that the compartmental model structure used in the MIP BAR model is simpler for the addition of this functionality, as well as being more easily extensible [26, 42].

## Q3—Combining Behaviour-based Aphid Rates with Multiple Infective Probes gives insights into how virus manipulation phenotypes affect aphid behaviour and virus transmission

**Virus manipulation of plant phenotype changes 'turnover' of aphid infective periods through its effects on aphid dispersal.** The more realistic aphid NPT virus transmission dynamics in our MIP-BAR model have implications for the effect of different virus phenotypes on epidemic outcomes. Maybe surprisingly, the effect of the variable aphid infectivity loss rate ($\widetilde{\tau}$) is mainly driven by the aphid dispersal rate ($\widetilde{\phi}$). $\widetilde{\tau}$ is directly proportional to $\widetilde{\phi}$, as in order to lose its infectivity an aphid must fly to a new plant and probe it (see Eq 19). This effect is most obvious for the 'attract-and-deter' scenario ($\nu > 1, \varepsilon < 1$), where despite the 'deter' phenotype suggesting that the virus remains in the aphid for longer by reducing the likelihood of feeding, there is actually an increased rate of virus loss from the aphids (so a shorter average infective period) as the epidemic progresses, driven by the increased dispersal rate between plants that feeding deterrence causes. This faster infectivity loss also represents the aphid more likely losing the virus from probing than feeding as it is less and less likely to feed as the proportion of infected plants increases. The increased rate of infectivity loss is paired with a

greatly increased rate of virus acquisition as the epidemic progresses, leading to a faster 'turn-over' of aphid infective periods, and a faster-growing and more severe epidemic (Fig 8). Donnelly et al. [17] also found that deterrence increased dispersal rate, leading to "more sustained feeding dispersals" later in an epidemic. However, this being linked to an increased turnover of aphid infectivity is to our knowledge a novel result. It is possible that this shorter infective period with increased dispersal rate could at least partly be an artefact of the model assuming instantaneous flight between plants. However aphids are generally only in appetitive flight between plants for minutes at a time [43], whereas aphids will often feed for several hours [41] (and we define average aphid feed length, $\eta = 0.83$ days). Therefore, incorporating flight time would likely only have a marginal effect on the length of the aphid infective period.

**Allowing for multiple infective probes increases the effectiveness of both virus manipulation scenarios, but in different ways.**   Given that an 'attract-and-deter' phenotype increases turnover of aphid infectivity, it therefore follows that decreasing the probability the aphid loses the virus when probing ($\rho$) reduces the degree of this turnover and hence reduces the increase in $\widetilde{\tau}$ over the course of the epidemic (Fig 8f). This further increases epidemic size and onset speed (Fig 8b) by causing a longer average aphid infective period for the same rate of dispersal and virus acquisition. The benefit of increased virus retention probability is only seen if the aphid rejects the plant; if it feeds it will not retain the virus anyway. In the case of a 'deter' phenotype, rejection is more likely on an infected plant. This offsets the fact that the addition of multiple infective aphid probes causes a bigger increase in virus retention probability for susceptible plants (Fig 7), resulting in a fairly consistent benefit of multiple infective probes across the epidemic (i.e. regardless of the ratio of infected to susceptible plants). Recent studies [17, 22] suggest that an attract-and-deter strategy is 'self-limiting'; deterrence increases risks of aphids not being able to settle and reproduce, as well as increasing risks of predation while flying between plants. This reduces the aphid population size [17] and hence limits larger-scale longer-term virus spread. There is also a well-reported limitation of attraction ($\nu > 1$) in that it increases initial disease spread but reduces final disease incidence [14, 17, 30], which our model confirms. The longer aphid infective period caused by multiple infective aphid probes could go some way to compensate for these limitations of an attract-and-deter phenotype—the risk aphids face from an increased dispersal rate due to deterrence, and the decrease in final epidemic size from attraction to infected plants—at least at a field scale.

For an 'attract-and-retain' phenotype, decreasing the probability that aphids lose infectivity when probing also increases the epidemic size and onset speed by increasing the average aphid infective period. However, in contrast to the 'attract-and-deter' scenario, this benefit is not consistent across the epidemic's progression: the addition of multiple infective aphid probes has the largest effect on aphid infective period at the start of the epidemic. This is because rejection (rather than feeding) of a plant is more likely for *susceptible* plants under a 'retain' phenotype, so benefitting from the increased virus retention during probing is more likely on a susceptible plant. Combined with the aforementioned larger increase in virus retention probability for susceptible plants (Fig 7) the result is that the aphid infective period is most increased when there are more susceptible plants, i.e. at the start of the epidemic (Fig 8f). This double effect of increasing aphid virus retention on susceptible plants also causes a larger increase in disease incidence for 'attract-and-retain' than 'attract-and-deter' (Fig 8h). However, adding multiple infective aphid probes conveys the largest increase to epidemic size and onset for phenotypes without an 'attract' component (Fig 8h); attraction to infected plants reduces the likelihood of landing on susceptible plants, decreasing the previously mentioned benefits of increased virus retention when probing them. Tungadi et al. [44] found a lack of 'attraction' ($\nu > 1$) phenotype of CMV-infected tobacco despite the presence of a 'retain' phenotype, and hypothesised this could reduce inhibition of transmission caused by the 'retain'

phenotype. A lower probability of aphid infectivity loss from probing ($\rho$) could have the same effect, counteracting the lower disease incidence caused by the 'retain' phenotype.

### Future directions and potential extensions

From our work, it is evident that the number of plants an aphid is able to probe while retaining its infectivity could greatly affect epidemic outcomes. For this reason, (1) experimental studies directly assessing the degree of virus loss from aphids' stylets during individual penetrations/intracellular punctures, and (2) field studies on the extent of virus spread possible from one infective aphid, would be valuable in assessing the contribution of the longevity of aphid infectivity to NPT virus transmission.

Our study looks at a single colonising aphid species. However, an 'attract-and-deter' strategy seems more likely for transient aphids that are not compatible with the host plant, and therefore will not settle down to feed [10]. Our model, along with the majority of previous NPT virus models [4, 17, 30] tracks one colonising aphid species, potentially making 'attract-and-deter' seem more self-limiting at the field-scale than it would be if the virus were spread by transient aphids passing through, that cannot settle on the host plant anyway. Additionally, $\rho < 1$ is probably more likely when the plant is a nonhost for the aphid, as very few brief exploratory probes would be done before rejection [3, 45–47]. It would therefore be interesting to adapt the MIP-BAR model to a transient aphid scenario. Although this has been explored in previous models [36, 48], these models do not include aphid probing/feeding behaviour at all.

Given the ease of extensibility of our MIP-BAR model's compartmental ODE structure [26, 42], it would also be useful to use the MIP-BAR model to re-examine previous analyses investigating other aspects of NPT virus transmission or virus manipulation of plant phenotype. Additionally, an interesting model extension would be to look at the possibility of using resistant 'trap' plants (that are attractive to aphids) as a potential NPT virus control mechanism, in a similar way to how 'push-pull' strategies have been employed for control of crop pests [34]. One limitation of our study is that, for ease of comparison between our model and previous models, we have not considered aphid population dynamics. This would therefore also be a useful model extension, because aphid feeding and settling behaviour and population size are linked. This is particularly relevant for the '(attract-and-)retain' phenotype, as it has been hypothesised that although this phenotype limits transmission on local time and spatial scales (which our model corroborates), it also encourages transmission on a larger landscape scale by encouraging aphid reproduction and hence production of more alate forms [17, 22]. We also note in passing that the MIP-BAR model technically needs a higher order correction term to account for inoculating the same plant multiple times in a single infective period [17]. However, we omitted this due to the marginality of its effect for anything other than very small plant population sizes.

### Conclusion

We demonstrate, through comparison of our MIP-BAR model to representative previous (MIP and BAR) models, that omission of either Behaviour-based Aphid Rates (BAR) or Multiple Infective Probes (MIP) affects epidemic outcomes, often causing underestimation of the potential magnitude of the effects of virus manipulation of plant phenotype on epidemic size. Firstly, Behaviour-based Aphid Rates, which are missing from the majority of models of NPT virus transmission (e.g. the MIP model), fundamentally change model behaviour and predictions by linking aphid dispersal and infectivity loss rates to both each other and to the stage of the epidemic. By comparing the MIP and MIP-BAR models (Figs 5 and 6), we show that inclusion of BAR particularly increases the advantage of the 'attract-and-deter' strategy for virus

transmission rates ([Fig 6]). We reveal for the first time this is likely due to faster aphid infective period 'turnover' caused by increased aphid dispersal ([Fig 8]). Secondly, by comparing the BAR and MIP-BAR models, we show the explicit Multiple Infective Probes functionality of our model increases the extent of virus transmission and hence final epidemic size, both when no virus manipulation is present, and under the virus manipulation scenarios ([Figs 7] and [8]). This is particularly apparent for the '(attract-and-)retain' plant phenotype, where aphids' ability to probe and infect multiple plants in one infective period counteracts the phenotype's self-limiting effects ([Fig 8]). However, the degree of ability of infective aphids to probe (and infect) multiple plants in one infective period is under-studied. Overall, our model allows better understanding of how virus manipulation of plant phenotype affects aphid behaviour and resulting aphid virus retention, thereby affecting virus disease epidemics. Our work also suggests new experimental and theoretical studies that should be done to further our understanding of NPT virus transmission, for better NPT virus prediction and control.

## Supporting information

**S1 Appendix. Relating the MIP-BAR and BAR models.** This appendix shows that the MIP-BAR model reduces to the BAR model when the aphid dynamics are held at pseudo-steady state, when $\rho = 1$ (i.e. without Multiple Infective Probes).
(PDF)

**S2 Appendix. Basic reproduction numbers for the three models.**
(PDF)

**S3 Appendix. With $\rho = 1$, the infectivity loss rate, $\widetilde{\tau}$, in the MIP-BAR model becomes equal to the aphid dispersal rate, $\widetilde{\phi}$, when $I = 0$.**
(PDF)

**S4 Appendix. Adding Multiple Infective Probes (MIP) to the BAR model is analytically intractable.**
(PDF)

## Acknowledgments

EKF and NJC would like to thank Trisna Tungadi, John Carr and Ruairi Donnelly for helpful discussions. EKF and NJC would also like to acknowledge Fred Fabre and Benoit Moury for making us aware of helpful literature.

## Author Contributions

**Conceptualization:** Elin K. Falla, Nik J. Cunniffe.

**Formal analysis:** Elin K. Falla.

**Investigation:** Elin K. Falla.

**Methodology:** Elin K. Falla, Nik J. Cunniffe.

**Software:** Elin K. Falla.

**Visualization:** Elin K. Falla, Nik J. Cunniffe.

**Writing – original draft:** Elin K. Falla, Nik J. Cunniffe.

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
