## [Decision Letter · Decision Letter 0]

15 Jul 2024

Dear Ms Falla,

Thank you very much for submitting your manuscript "Why aphid virus retention needs more attention: modelling aphid behaviour and virus manipulation in non-persistent plant virus transmission" for consideration at PLOS Computational Biology. As with all papers reviewed by the journal, your manuscript was reviewed by members of the editorial board and by several independent reviewers. The reviewers appreciated the attention to an important topic. Based on the reviews, we are likely to accept this manuscript for publication, providing that you modify the manuscript according to the review recommendations.

Sincerely,

Claudio José Struchiner, M.D., Sc.D.

Academic Editor

PLOS Computational Biology

Virginia Pitzer

Section Editor

PLOS Computational Biology

Reviewer's Responses to Questions

**Comments to the Authors:**

Reviewer #1: The authors developed and analyzed models for a very important system: plant epidemics caused by viruses that are non-persistently transmitted (NPT) by insect vectors (primarily, aphids). Plant viruses cause numerous serious diseases of all crops, with epidemics often resulting in large economic losses. There are several classes of plant virus transmission by insects, and NPT may be the most common. There are unique properties of the NPT class, especially related to the high rate at which vectors acquire the virus via probing of plants (to determine if the plant is a suitable feeding host species or variety), and then the high rate at which vectors lose infectivity through feeding from plants. There have been models previously developed to better understand epidemics caused by plant viruses, especially compartmental (ODE) ones, and the authors do an excellent job of describing the relevant models. As stated here, these previous models do not take into account sufficient aspects of virus effects on vector behavior (mediated through the host plant) that can have a potentially large impact on epidemics (such as plant epidemic “size”, invasion thresholds, or rates of increase in disease).

The current manuscript builds nicely on the ODE compartmental model developed by Madden, Jeger, and van den Bosch over 20 years ago, simplifying some components that are not overly relevant for the special NPT class, and elegantly expanding other components to represent insect behavior as affected by the virus. They compare this to a Monte Carlo equilibrium model that Cunniffe and colleagues developed that distinguishes insect probing versus feeding, and deals with vector dispersal and loss of infectivity on plants. They then developed a new (third) ODE model that essentially incorporates some vector-behavior features of the Monte Carlo model into their ODE model. This third model is known as the MIP-BAR model. The authors do an excellent job of explaining the development of the models, the meaning of the parameters, and results obtained through analysis of the model equations and with simulation. The models are presented in an intuitive way, and key epidemic scenarios are shown with clear graphic results.

Of major importance is their ingenious approach to parameterizing plant attractiveness to vectors and plant acceptability once probing of the plant occurs. These are key aspects of the virus-induced vector phenotypes of “attract-and-deter” and “attract-and-retain”. These behaviors have been documented in recent biological studies, and results from the modelling work here show the large impact of these behaviors on plant epidemics. This is highly relevant in (ultimately) developing and testing strategies for epidemic control, although control is not part of this manuscript. Also ingenious is the derivation of a variable vector dispersal rate and the variable loss of infectivity term.

An additional significant contribution is their derivation of the basic reproduction number, Ro, using the ‘next generation’ method (which I consider to be the best approach for this derivation). They explain their Ro terms in a clear and logical manner.

The authors appropriately discuss ways that the modeling will be extended in future work. One scenario mentioned on page 21 deals with the consideration of transitory vectors rather than colonizing vectors. I hope the authors do work on this since it is common for plant epidemics to occur due to the transmission of NPT viruses by transitory (non-colonizing) vector species, often multiple different aphid species moving through fields at the same time (where none of the aphid species colonize the plants). This might be a good place to cite Marcus & Raccah (1986; J. Appl. Stat. 13: 167-175). I know this is ancient history (!), but these authors do develop a simple discrete-time model for NTP viruses, where there is a combination of colonizing and transitory vectors. Although crude compared to the new work in the current manuscript, I do see relevance for the discussion.

Reviewer #2: This manuscript explores whether previous approaches to model vector-borne diseases caused by non-persistent viruses (NPV) in plants can be unified in a single framework. More specifically, most ODE models (e.g., Hamelin et al 2023) account for vector preferences but ignore other NPV specifics, including the distinction between probing and feeding (Cunniffe et al 2021 is an exception). Probing may be sufficient to acquire and transmit the virus, but the vector may decide to stay longer and feed, depending on whether the plant is infected or not. In addition, virus loss is more likely on susceptible plants compared to infected plants (in which reacquisition of the virus can occur). Therefore, the rate at which vectors lose the virus and the number of plant visits per unit time (termed the “dispersal rate” in this study, even though the model is nonspatial) are dependent on S and I (the densities of susceptible and infected hosts), which is not accounted for in traditional ODE models. Donnelly et al (2017) tackled this issue in an original way, using Markov Chain theory. However, their model assumed fast vector infection dynamics relative to plant infection dynamics, which makes sense, but is less general than Cunniffe et al (2021), who additionally considered conditional vector preferences and vector population dynamics. The model presented in this study bridges the gap between Donnelly et al (2017) and Cunniffe et al (2021). The main message is that NPV specifics matter, and can be accounted for in a relatively simple ODE model. Overall, this work constitutes a useful contribution to the field of mathematical and computational plant disease epidemiology.

The manuscript is very well written, and contains many results/figures (including compelling mathematical results in appendix). My comments below intend to improve their presentation.

- In the Abstract, the sentence starting with “Additionally” sounds contradictory with the previous sentence. I would suggest replacing “Additionally” with “However” and deleting “However” from the next sentence.

- L15: I would suggest replacing “subsequent” with “uninfected” since reacquisition of the virus can occur on infected plants.

- L132-135: you state that “infective aphid feeding on a plant will always cause it to lose infectivity”. This sounds like a very strong assumption when the plant is infected. Cannot the aphid acquire the virus again when feeding? I would assume the sap the aphid feeds on also contains viral particles, doesn’t it? Similarly, you state “in order for virus acquisition to occur the aphid must reject (i.e., not feed on)” the plant host. I think these assumptions should be more explicitly justified, right from the Introduction (around L15 and reference [8,9]), since they are key to understand the model then.

- L153: in the Introduction, you made the point that “attract and retain” could be a way for the virus to increase its transmission provided “retain” means the growth of a local colony and the production of winged individual when the local density is high enough. But then you assume the aphid population is a constant, and is composed of winged individuals only. What is the point of considering “retain” in this model then?

- L193: I would have appreciated a rederivation of Eq. (9), in a box or at least in an appendix, for the manuscript self-sufficiency, especially since the Markov Chain is alluded to in Fig. 2c. I realize that the appendix of Donnelly et al (2017) may not be that amenable to simplification though. Would it be possible to provide the reader with an intuition about the interpretation of Eq. (9)?

- L199: Ideally, I would have liked the authors to present the most general (termed “MIP-BAR”) model first, and then show how it simplifies to previous subcases (MIP) and (BAR). The authors do a good job at showing that BAR corresponds to a slow/fast approximation of MIP-BAR, but whether MIP can be viewed as a special case of MIP-BAR is left implicit. I understand that this is not the case (due to the epsilon factor associated with v among other things), but this might be clarified.

- L592-595: I wonder whether this observation could be an artifact due to the fact that the travel time between plants is assumed to be zero. I would appreciate a discussion around this assumption (i.e., the “shorter average infective period” does not account for travel times, so it should not be interpreted in a straightforward way).

A more general comment has to do with the fact that many results are presented, some being more expected/intuitive than others. It is difficult to quickly grasp the most salient results from the manuscript in its current form. Please consider a way to improve this.

**Have the authors made all data and (if applicable) computational code underlying the findings in their manuscript fully available?**

Reviewer #1: Yes

Reviewer #2: Yes

PLOS authors have the option to publish the peer review history of their article (what does this mean?). If published, this will include your full peer review and any attached files.

Reviewer #1: No

Reviewer #2: No

Figure Files:

Data Requirements:

Reproducibility:

References:

---

## [Editor Report · Decision Letter 1]

11 Sep 2024

Dear Ms Falla,

We are pleased to inform you that your manuscript 'Why aphid virus retention needs more attention: modelling aphid behaviour and virus manipulation in non-persistent plant virus transmission' has been provisionally accepted for publication in PLOS Computational Biology.

Best regards,

Claudio José Struchiner, M.D., Sc.D.

Academic Editor

PLOS Computational Biology

Virginia Pitzer

Section Editor

PLOS Computational Biology

---

## [Editor Report · Acceptance letter]

23 Sep 2024

PCOMPBIOL-D-24-00836R1 

Why aphid virus retention needs more attention: modelling aphid behaviour and virus manipulation in non-persistent plant virus transmission

Dear Dr Falla,

I am pleased to inform you that your manuscript has been formally accepted for publication in PLOS Computational Biology. Your manuscript is now with our production department and you will be notified of the publication date in due course.

With kind regards,

Anita Estes
